# Unraveling the molecular architecture of autoimmune thyroid diseases at spatial resolution

Rebeca Martínez-Hernández [1,6] ✉, Nuria Sánchez de la Blanca [1,6], Pablo Sacristán-Gómez[1,6], Ana Serrano-Somavilla[1], José Luis Muñoz De Nova[2], Fátima Sánchez Cabo [3], Holger Heyn [4,5], Miguel Sampedro-Núñez[1] & Mónica Marazuela[1] ✉

Autoimmune thyroid diseases (AITD) such as Graves' disease (GD) or Hashimoto's thyroiditis (HT) are organ-specific diseases that involve complex interactions between distinct components of thyroid tissue. Here, we use spatial transcriptomics to explore the molecular architecture, heterogeneity and location of different cells present in the thyroid tissue, including thyroid follicular cells (TFCs), stromal cells such as fibroblasts, endothelial cells, and thyroid infiltrating lymphocytes. We identify damaged antigen-presenting TFCs with upregulated CD74 and MIF expression in thyroid samples from AITD patients. Furthermore, we discern two main fibroblast subpopulations in the connective tissue including ADIRF+ myofibroblasts, mainly enriched in GD, and inflammatory fibroblasts, enriched in HT patients. We also demonstrate an increase of fenestrated PLVAP+ vessels in AITD, especially in GD. Our data unveil stromal and thyroid epithelial cell subpopulations that could play a role in the pathogenesis of AITD.

The prevalence of autoimmune thyroid diseases (AITD) has increased in recent years, reaching 5% of the population worldwide. AITD are complex diseases that result from a dysregulation of the immune response against thyroid antigens. AITD are classified in two major categories based on their clinical outcomes and immune phenotype: Hashimoto´s thyroiditis (HT) and Graves´ disease (GD)[1,2]. Both are characterized by the presence of circulating thyroid antibodies and infiltration by autoreactive lymphocytes of the thyroid gland, and sometimes the orbit (Thyroid eye disease, TED). In HT, the breakdown of immune tolerance results in thyroid destruction and hypothyroidism in most cases[3,4]. Conversely, GD is characterized by thyroid follicular hyperplasia and a hyperthyroidism secondary to the

effect of antibodies against the thyroid stimulating hormone (TSH) receptor (TSH-R)[5].

From the pathological point of view, HT exhibits a rich lympho-cytic infiltrate commonly located in follicles with germinal center formation[6]. Lymphocytes are predominantly T cells along with numerous B lymphocytes and macrophages polarized to an inflammatory phenotype. Thyroid follicles are usually atrophic and outlined by cuboidal cells. As the disease progresses, the excessive accumulation of collagen fibers may lead to fibrosis in the interstitial tissue[7,8]. GD shows a variable degree of lymphocytic infiltrates as well as lymphoid follicles. The major difference with HT is the follicular architecture and cytomorphology of the lining cells; in GD, hyperplastic follicles present

[1]Department of Endocrinology and Nutrition Hospital Universitario de la Princesa, Instituto de Investigación Sanitaria Princesa, Universidad Autónoma de Madrid, and Centro de Investigación Biomédica en Red de Enfermedades Raras (CIBERER GCV14/ER/12), Madrid, Spain. [2]Department of General and Digestive Surgery, Hospital Universitario de la Princesa, Instituto de Investigación Sanitaria Princesa, Universidad Autónoma de Madrid, Madrid, Spain. [3]Bioinformatics Unit, Centro Nacional de Investigaciones Cardiovasculares (CNIC), Madrid, Spain. [4]Centro Nacional de Análisis Genómico (CNAG), Barcelona, Spain. [5]Universitat de Barcelona (UB), Barcelona, Spain. [6]These authors contributed equally: Rebeca Martínez-Hernández, Nuria Sánchez de la Blanca, Pablo Sacristán-Gómez. ✉e-mail: rebeca.martinez@salud.madrid.org; monica.marazuela@uam.es

papillary epithelial infoldings and the thyroid epithelium is lined by columnar cells[8,9].

The pathogenesis of AITD is probably related to a complex and multifactorial interplay of specific susceptibility genes and environmental exposures, leading to the breakdown of self-tolerance and subsequent anti-thyroid autoimmune responses[10–13]. As an autoimmune disease, the disturbance of the tolerance against self-antigens is the main pathogenic mechanism with a large cellular interplay that involves TFCs, stromal cells and lymphocytes. Several studies have focused on the different immune subsets that are probably involved in the pathogenesis of AITD[14–20].

However, beyond the clear contribution of immune cells to AITD development, elements of the own thyroid tissue, such as thyroid follicular cells (TFCs) and stromal cells, including fibroblasts and endothelial cells, have been less studied. TFCs contribute to the presentation of thyroid antigens to immune cells via major histocompatibility complex II (MHC-II) molecules and play an important role in regulating intrathyroidal T cell survival and selection[21,22]. Fibroblasts might have a role in inflammation leading to cytokine production and in tissue repair. Indeed, fibroblasts affected by the proinflammatory environment present in AITD can undergo a transition to myofibroblasts[23,24]. Endothelial cells experience an increase in permeability that facilitates chemotaxis of immune cells and their infiltration in thyroid tissue[25–27]. Considering the large cellular interplay in AITD and the limited information published about epithelial and stromal thyroid cells, a further understanding of the roles of these cells in the pathogenesis of AITD will contribute to clarify the mechanism of these diseases and develop novel therapeutic targets.

Here, we use spatial transcriptomics (ST) to better understand the cellular specificity and complexity of different cell subsets involved in AITD and their possible pathogenic role. Histological classification was applied to identify the differences and similarities of cell states between AITD and control tissues using spatial information. This approach enabled the characterization of cellular and molecular signatures in AITD, including the diversity of TFCs, fibroblasts, vessels, and thyroid infiltrating lymphocytes (TILs), and how they distribute in the autoimmune thyroid tissue. In addition, we deepen our knowledge on several pathways and mechanisms that could potentially contribute to the pathogenesis of these disorders (depicted in Fig. 1a).

## Results

### Quality control and first analysis

We profiled and sequenced spatial gene expression from eight thyroid tissue samples (three HT: HT1, HT2, HT3; three GD: GD1, GD2, GD3, and two controls: C1 and C2). We sequenced each sample with a mean of 84602 ± 38060.11 reads under spots, with a median of 1197 (Q1:740.750-Q3:1792.625) genes per spot. Number of counts, number of features, and percentage of ribosomal and mitochondrial genes were similar between samples (Supplementary Fig. 2a). 407 spots were removed from the HT3 sample due to artificial scratches that correlated with low quality measures (Supplementary Fig. 2b).

Single cells from HT samples were analyzed to identify the cell populations reported by ref. 27. After the removal of doublets, we analyzed 27,546 cells. Then, we performed a sample integration with harmony to remove batch effects. At 0.3 resolution, we found 11 major clusters annotated based on their top markers (Supplementary Fig. 3a, b). Correlation with published profiles is represented in Supplementary Fig. 3c.

### Analysis of histology-based annotation profiling of ST data

To determine purer cell-type signatures within the tissue, we first annotated each spot based on its histological pattern in the hematoxylin and eosin image (Supplementary Fig. 1). The histologic annotation identified eight regions according to the predominant cell tissue: TFCs, thyrocytes from nodule (Nodule TFCs), TILs, germinal

centers (GC), connective tissue (CT), vessels (V), colloid-macrophages (Col) and Borderline (BL). Borderline margins were assigned in those areas between TFCs and TILs (Fig. 1b). The results, as expected, showed a diminished representation of thyrocyte spots and a high presence of immune infiltration and germinal centers in HT samples (Fig. 1b).

When we evaluated the performance of histological classification and its reliability using the molecular data, we first integrated the eight samples using Harmony[28]. UMAP plot showed a correct merge of the spots and a divergence based on the predominant cell population (Fig. 1c). Results from testing the distribution of the samples and conditions in the UMAP suggested an optimal integration, in which TILs and GC were exclusively present in AITD spots and then separated from TFCs and stroma, as expected (Fig. 1d, e). Next, each group was profiled by different canonical markers such as: TG, TPO, and TSHR for TFCs and Nodule (TFCs); CD3D, T Cell Receptor Alpha Constant (TRAC) for TILs; Follicular Dendritic Cell Secreted Protein (FDCSP) and C-X-C Motif Chemokine Ligand 13 (CXCL13) for GC; CD34 and vimentin (VIM) for CT; Von Willebrand Factor (VWF) and Myosin Heavy Chain 11 (MYH11) for V and finally lactotransferrin (LTF) and CD68 for Col. These molecular markers confirmed the correct classification of the histological pattern (Fig. 1f). Supplementary Data 1 lists the top marker genes for each of the seven groups already described.

To confirm the unbiased groups for the histology-based approach, we also performed a clustering analysis of the integrated object using the eight samples at 0.1, 0.3, and 0.5 resolution (Supplementary Fig. 4). Regarding the similarities between both approaches, we opted for histological annotation instead of unsupervised analysis to prevent mixed spots and borderline areas.

To further increase our knowledge of the stromal microenvironment in AITD pathogenesis, we specifically analyzed four of the major compartments comprising TFCs, connective tissue, vessels, and TILs separately, to depict molecular differences between conditions.

### Molecular profiling of TFCs revealed altered subpopulations of TFCs in AITD

First, we focused on TFC state using the Thyrocyte Differentiation Score (TDS). This signature quantifies their differentiation grade by averaging the expression of TG, TPO, Solute Carrier Family 26 Member 4 (SLC26A4), Iodothyronine Deiodinase 2 (DIO2), TSHR, Paired Box (PAX8), Dual Oxidase 1 (DUOX1), Dual Oxidase 2 (DUOX2), NK2 homeobox 1 (NKX2-1), GLIS Family Zinc Finger 3 (GLIS3), Forkhead Box E1 (FOXE1), Trefoil Factor (TFF3) and Four-And-A-Half LIM Domains 1 (FHL1) genes[29,30] through AddModuleScore function (Supplementary Data 2). When we statistically analyzed differences in TDS between the three conditions, we found that HT TFCs (median = −0.009 [−0.154, 0.142]) were poorly differentiated compared to control and GD samples (median = 0.785 [0.576, 0.975] and 0.755 [0.539, 0.958], respectively, p adjusted < 0.001). Interestingly, controls had slightly higher TDS than GD samples (p adjusted = 0.006) (Fig. 2a). Spatial visualization of the TDS revealed a differential distribution, with lower values observed in TFCs spots closer to the immune infiltrates (Fig. 2b). Further validation in single cell data from HT thyroids corroborated the existence of two thyrocyte populations (Supplementary Fig. 5a) with high and low TDS (Supplementary Fig. 5b).

To study the molecular differences between conditions, we performed an unsupervised clustering from the integration of the ST spots corresponding to TFCs regions. At a resolution of 0.5 we found six different clusters (Fig. 2c). Cluster T1 (Fig. 2c, d cluster in red) was almost exclusively represented in HT and GD spots (AITD TFCs) (Fig. 2e). Concerning the other clusters, TFC markers such as TG, TFF3, PAX8, and SLC26A7 were upregulated in T0 and T2 (TFC1 and TFC2); mitochondrial genes (i.e., NQO1, NDUFB2 and UQCR11) were upregulated in T3 (High metabolic TFCs, TFC3); some TFC and blood markers were upregulated in T4 (TFC + blood) and CT-related genes

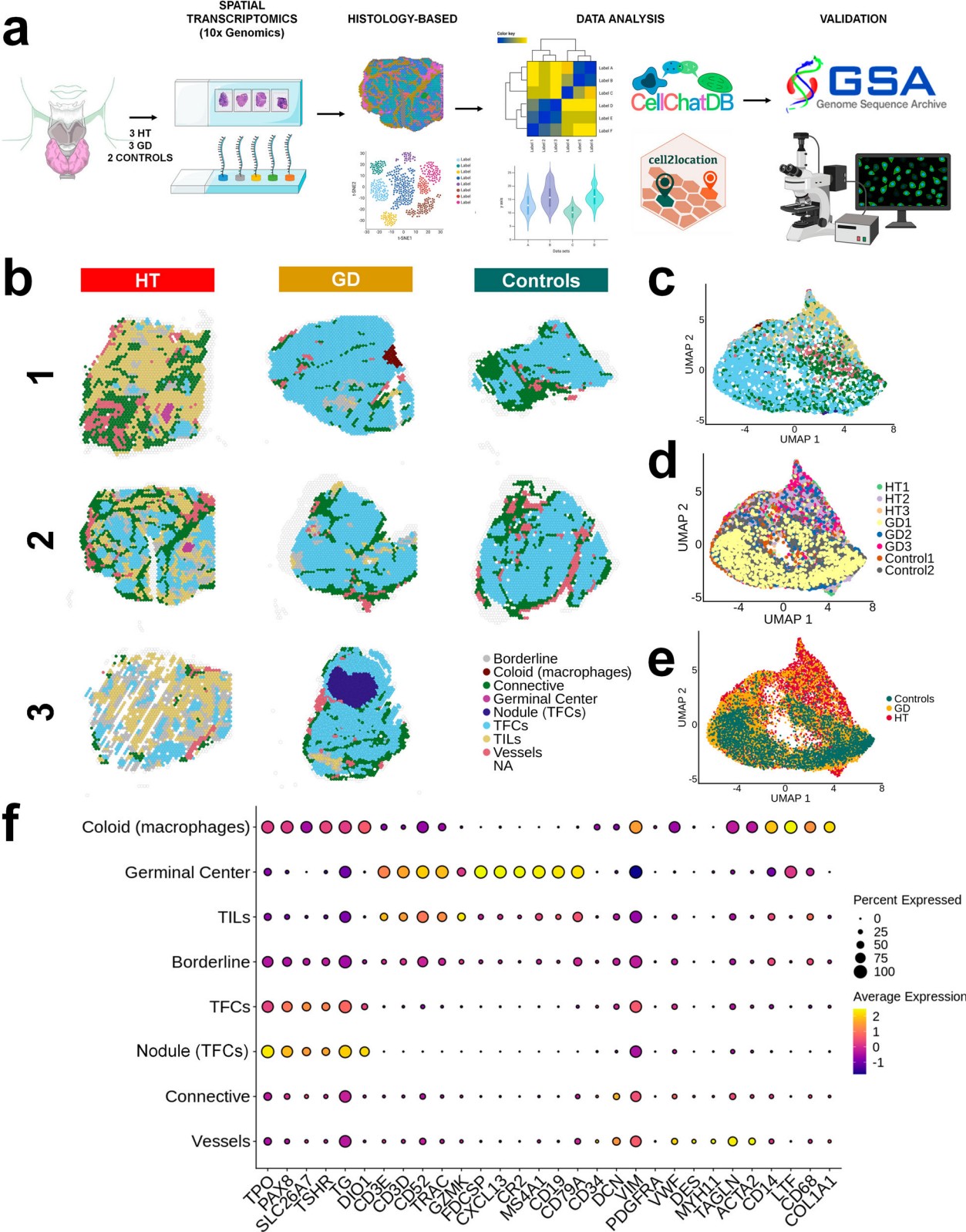

**Fig. 1 | Histology-based classification. a** Overview of the study design. Created with BioRender.com, released under a Creative Commons Attribution-NonCommercial-NoDerivs 4.0 International license. **b** Spatial transcriptomics spots were classified according to their histological appearance. **c** Molecular UMAP colored according to histology-based classification. **d** Molecular UMAP of samples.

**e** UMAP colored according to the different conditions. **f** Dot plot of canonical markers per tissue and their expression across areas. TFCs thyroid follicular cells, TILs thyroid infiltrating lymphocytes, HT Hashimoto´s thyroiditis, GD Graves´ disease, UMAP Uniform Manifold Approximation and Projection for Dimension Reduction.

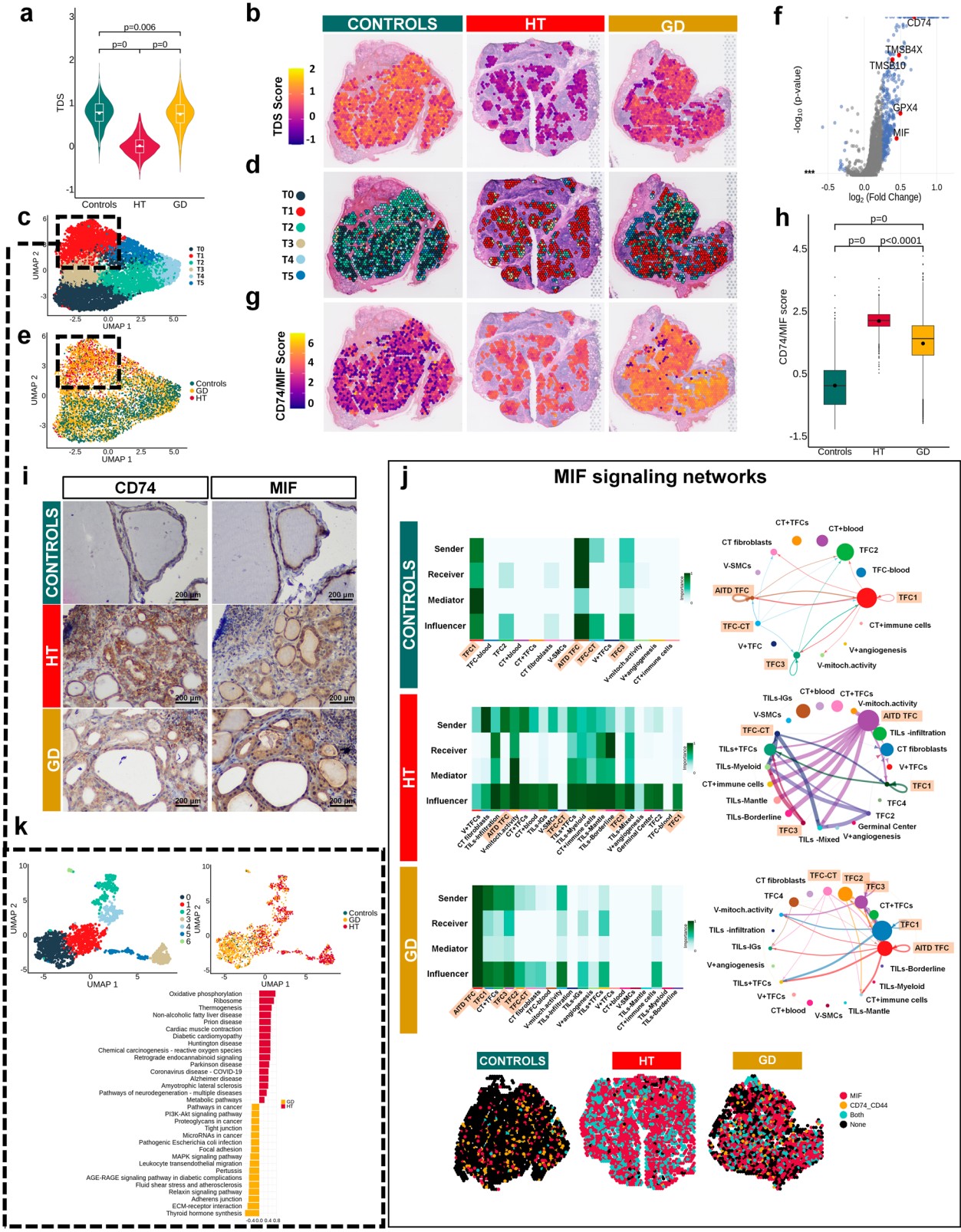

such as *DCN* and *FBLN1* were upregulated in T5 (TFC-CT) (Supplementary Data 3).

Differential expression (DE) analysis between cluster T1 and the rest of the clusters showed the presence of immunoglobulins, probably due to a cross-reaction between TFCs and plasma cells located in the proximity (Supplementary Data 4, Supplementary Fig. 6a, b). Interestingly, this analysis highlighted the increased expression in

TFCs of selenoprotein genes such as Glutathione Peroxidase 4 (*GPX4*), which are important for the correct removal of reactive oxygen species in the thyroid gland[31], as well as different thymosins such as Thymosin Beta-4 (*TMSB4X*) and Thymosin Beta-10 (*TMSB10*), which contribute to cytoskeleton regulation and are upregulated in malignant cells[30,32] (Fig. 2f). Furthermore, ligand-receptor analysis confirmed that the number of interactions involving this AITD-associated TFCs

**Fig. 2 | Molecular profiling of TFCs identifies altered subpopulations in HT and GD samples. a** Violin plots showing the comparison of Thyrocyte Differentiation Score (TDS) of TFCs spots between conditions. HT ($n = 3$): spots = 1247, median = −0.009 [−0.154, 0.142]; GD ($n = 3$): spots = 4133, 0.755 [0.539, 0.958]; controls ($n = 2$): spots = 1884, median = controls: 0.785 [0.576, 0.975]; all adjusted two-sided $p$ values. **b** Spatial distribution of the TDS in TFCs spots. **c** UMAP at 0.3 resolution of TFCs spots. **d** Spatial disposition of TFCs clusters at 0.3 resolution. **e** UMAP of TFCs spots colored according to conditions: controls, GD, and HT. **f** Volcano Plot of differential expression analysis between cluster T1 (red) and the rest of the clusters. Blue spots represent statistically significant genes with logarithmic fold changes of more than +0.25 and less than −0.25. **g** Spatial distribution of the *CD74-MIF* score in TFCs spots. **h** Box plot of the *CD74* and *MIF* score in all conditions. HT ($n = 3$): spots = 1247, median = 2.202 [2.016, 2.394]; GD ($n = 3$): spots = 4133, median = 1.620 [1.092, 2.036]; controls ($n = 2$): spots = 1884, median = 0.115 [−0.485, 0.605]; all adjusted two-sided $p$ values. **i** Representative images of CD74 and MIF

immunostaining in serial sections of thyroid tissue from controls, HT, and GD patients. Scale bar: 200 µm. Stainings were confirmed in at least seven biological replicates. **j** *MIF* signaling networks: Heatmaps display the condition of each spatial cluster as sender or receiver of *MIF* ligand and their contribution. Circle plot represents the strengthening of the TFCs clusters as senders of MIF and their receivers. Spatial plot of *MIF* and *CD74-CD44* in controls, HT, and GD samples. **k** Re-clustering of the AITD TFCs cluster (T1). Top left: UMAP of the re-clustering of the AITD TFCs cluster. Top right: Re-clustering colored by conditions. Below: Barplot of the top KEGG pathways from the differential gene expression between HT (dark red) and GD (dark yellow) in the re-clustering. Dunn test for statistical analysis and adjusted two-sided $p$ values with Holm (**a**, **h**) and with Bonferroni (**f**). TDS thyroid differentiation score, HT Hashimoto´s thyroiditis, GD Graves´ disease, UMAP Uniform Manifold Approximation and Projection for Dimension Reduction, TFCs thyroid follicular cells, CT connective tissue, V Vessels, TILs thyroid infiltrating lymphocytes, SMCs smooth muscle cells, IGs immunoglobulins.

cluster was higher than those interactions involving the other clusters in the TFCs region (Supplementary Fig. 7). When we performed a biological enrichment analysis, we showed an increased presence of autoimmune-related pathways and antigen presentation related molecules in this cluster (Supplementary Data 4). Interestingly, we observed an upregulation of the *CD74*/Macrophage migration inhibitory factor (*MIF*) system on TFCs which has been related to injury repair and antigen presentation pathways (Fig. 2f, g, Supplementary Data 4). Indeed, when we statistically analyzed differences of *CD74*/*MIF* co-expression within the TFCs spots between conditions, we found significantly higher levels of *CD74*/*MIF* in HT (median = 2.202 [2.016, 2.394]) and GD (median = 1.620 [1.092, 2.036]) compared to controls (median = 0.115 [−0.485, 0.605], $p$ adjusted < 0.0001) (Fig. 2h). Spatial plotting focused on TFCs exposed a differential distribution of *CD74*/MIF co-expression which was higher in areas close to the immune infiltrate, similarly to TDS (Spearman rho test: $r = −0.44$, $p < 0.001$), suggesting again two subpopulations differentiated by their proximity to TILs (Fig. 2g). To further validate this presumption, we plotted *CD74*/*MIF* on TFCs from HT single cell data confirming that TFCs with low TDS tended to upregulate *CD74* and *MIF* (Supplementary Fig. 5c). Immunostaining of CD74 confirmed the increased expression of this receptor on local TFCs close to infiltrating lymphocytes in HT and GD, reflecting a differential TFCs landscape in these conditions. On the other hand, MIF was expressed in the cytoplasm of both AITD and control TFCs (Fig. 2i and Supplementary Fig. 8).

We then attempted to confirm the active involvement of *CD74*/*MIF* TFCs in the pathogenesis of AITD by conducting cell-cell interaction analysis within our ST data using CellChat[33]. The canonical *CD74-MIF* interaction requires the participation of *CD44* as a co-receptor. This analysis identified AITD-associated spots from HT samples as *MIF* senders, which interacted with immune clusters. However, these TFCs were not *MIF* receivers (Fig. 2j). Indeed, AITD-associated TFCs showed increased interactions with immune cells related to antigen presentation pathways (*HLA* or *MHC-II*), suggesting an alternative role of *CD74* beyond the *CD74-CD44*/*MIF* axis (Supplementary Fig. 9). On the other hand, AITD associated spots from GD samples were both MIF senders and receivers, with an increased communication probability for the *CD74-CD44*/*MIF* interaction compared to controls (Fig. 2j). Moreover, these TFCs did not present interactions with immune cells related to antigen presentation (Supplementary Fig. 9). Interestingly, both conditions exhibited *MIF*/*CD74-CXCR4* interactions, considering AITD-associated spots as MIF senders and immune cell clusters as *CD74-CXCR4* receptors (Fig. 2j and Supplementary Fig. 9). This interaction is probably related to T cell chemotaxis, as previously reported[34].

To identify differential thyroid subpopulations between both diseases, we performed a subsequent re-clustering of this AITD-associated cluster. We obtained seven clusters at 0.3 resolution

(Fig. 2k). Notably, we observed a clear separation between HT and GD spots into different clusters: cluster 0 was mostly associated with GD; clusters 2–6 associated with HT, while both conditions overlapped in cluster 1 (Fig. 2k). Regarding the differences observed between HT and GD TFCs, we performed a DE analysis between both conditions. The enrichment analysis of the DE genes in GD thyrocytes showed an increased representation of pathways related to extracellular matrix, cell junctions, cell adhesion and thyroid hormone synthesis (Fig. 2k). On the other hand, in HT there was an increase of reactive oxygen species, electron transport chain (ETC), oxidative phosphorylation and pathways associated with oxidative stress-derived damage (i.e., Huntington, Parkinson, Alzheimer diseases and amyotrophic lateral sclerosis) compared to GD (Fig. 2k). Moreover, we performed a DE followed by an enrichment analysis of TFCs pseudobulk area confirming the pathways associated to AITD and the differences between both diseases already described (Supplementary Fig. 10).

In concordance with ST results, these molecular signatures associated with ETC and stress/damaging pathways (Supplementary Data 2) were also confirmed with scRNA-seq data from HT. Interestingly, both were enhanced in thyrocytes characterized by a low differentiation state and an upregulation of CD74/MIF (Supplementary Fig. 5d, e). When we performed the deconvolution of HT spots, "damaged" TFCs were present in higher proportion than"non-damaged" TFCs in all clusters (Supplementary Fig. 6a).

Overall, we identified a subpopulation of thyrocytes in AITD that express the CD74/MIF axis close to the immune infiltrate. In HT, this subpopulation exhibited a poor differentiated signature and damaged-associated pathways.

### Connective tissue profiling unveiled specific fibroblast signatures: GD-associated ADIRF+ myofibroblasts in interfollicular areas and HT inflammation-associated fibroblasts (IAFs) in connective tissue areas

To further address the transcriptional patterns of the connective tissue, spots were first clustered by unsupervised analysis. At 0.3 resolution, we obtained five clusters (Fig. 3a, b; C0-C4). We observed a high expression of thyroglobulin, probably due to an overlap of connective tissue and TFCs (Supplementary Fig. 6a). Regarding the top markers in each cluster, we observed that cluster C0 showed less *TG* content and was mostly expressed in HT and GD (Fig. 3a–c; Supplementary Fig. 6a); whereas subgroup C1 expressed high levels of hemoglobin genes (*HBB, HBA1, HBA2*), clusters C2 and C3 expressed thyrocyte-associated markers (i.e., *TG, DIO2, PAX8, TPO*) and C4 exhibited a high presence of immune cell-related genes (i.e., *FDCSP, CD52, CD79A*) (Fig. 3a–c, Supplementary Data 5).

To obtain the exclusive molecular signatures of disease-associated connective tissue, we selected cluster C0 to perform a hierarchical clustering of its top 40 marker genes, excluding ribosomal protein related genes (Fig. 3d). We applied functional annotation of

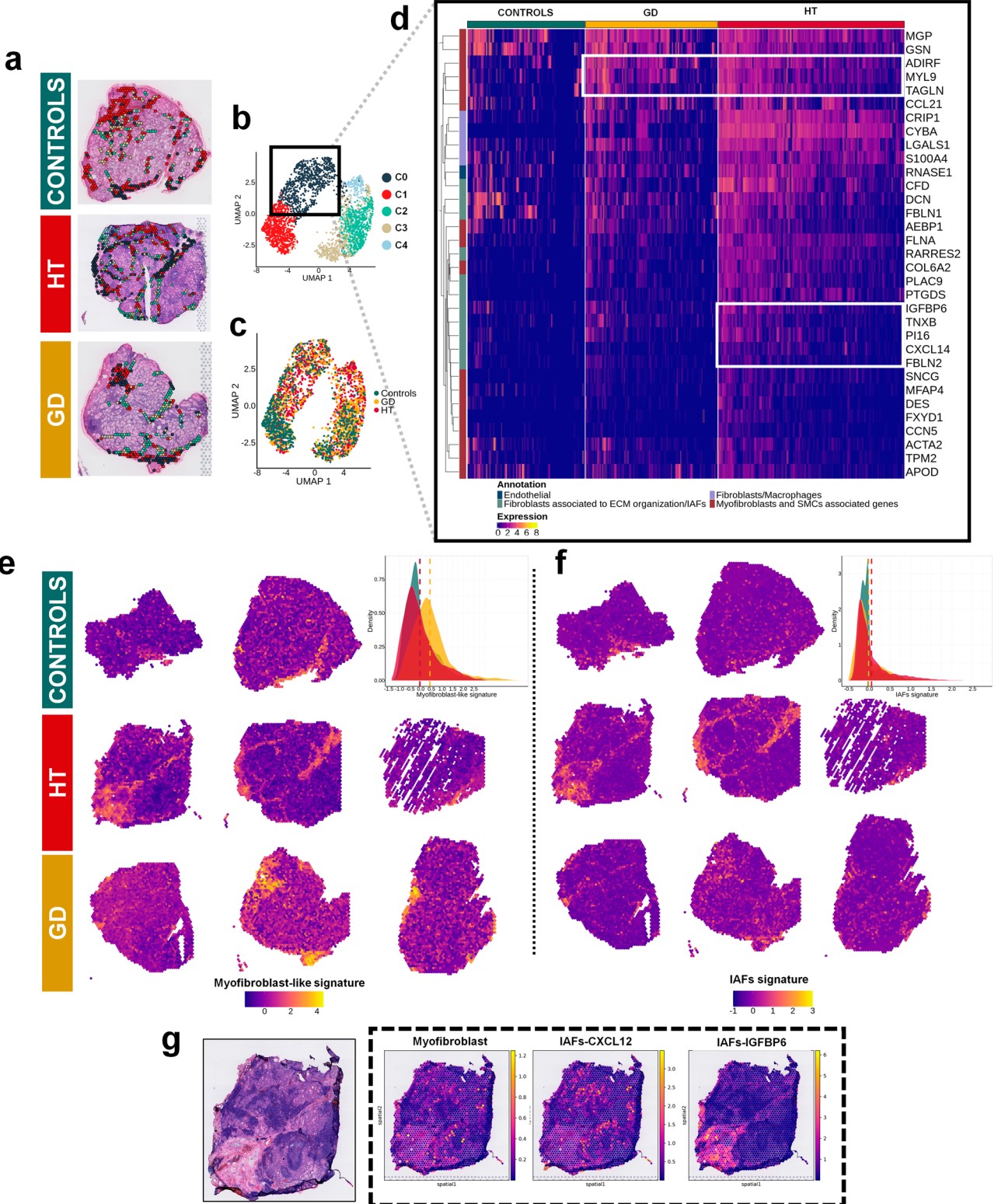

**Fig. 3 | Analysis of the connective tissue reveals different fibroblast subpopulations in HT and GD samples. a** Spatial distribution of CT clusters at 0.3 resolution. **b** UMAP of the connective tissue clustering at 0.3 resolution. **c** Distribution per condition in the UMAP colored by condition. **d** Hierarchical clustering of the top markers of cluster 0. **e** Myofibroblasts-like signature: Spatial distribution and density plot across conditions. HT ($n = 3$): spots = 6403, mean = −0.046 ± 0.770, GD ($n = 3$): spots = 6583, mean = 0.431 ± 0.844, controls ($n = 2$): spots = 3592, mean = −0.035 ± 0.725. **f** IAFs signature: Spatial distribution and density plot across conditions. HT ($n = 3$): spots = 6403, mean = 0.051 ± 0.429, GD

($n = 3$): spots = 6583, mean = −0.023 ± 0.362, controls ($n = 2$): spots = 3592, mean = −0.033 ± 0.280. **g** Spots deconvolution of HT samples: proportion and location of myofibroblasts and IAFs subpopulations (*CXCL12* IAFs and *IGFBP6* IAFs). Side bars show the expected cell abundance. Dunn test for statistical analysis and adjusted two-sided *p* values with Holm (**e, f**). IAFs inflammatory-associated fibroblasts, HT Hashimoto´s thyroiditis, GD Graves´ disease, UMAP Uniform Manifold Approximation and Projection for Dimension Reduction, ECM extracellular matrix, SMCs smooth muscle cells.

each gene based on its molecular expression into different categories: endothelial, fibroblasts associated to extracellular matrix (ECM) organization/inflammatory associated fibroblasts (IAFs), fibroblasts/macrophages, and myofibroblast and smooth muscle cells (SMCs) associated genes (Fig. 3d).

Interestingly, we observed two signatures distributing disparately that clustered together across conditions (Fig. 3d): (i) The first subset of genes enhanced in HT and GD included Transgelin (*TAGLN*), Myosin Light Chain 9 (*MYL9*) and the Adipogenesis Regulatory Factor (*ADIRF*), which were attributed to contractile cells, such as myofibroblast-like cells (myofibroblasts or pericytes) or SMCs. (ii) The second group of genes (mostly represented in HT samples) included the Insulin Like Growth Factor Binding Protein 6 (*IGFBP6*), Tenascin XB (*TNXB*), Peptidase Inhibitor 16 (*PI16*), C-X-C Motif Chemokine Ligand 14 (*CXCL14*) and Fibulin 2 (*FBLN2*), genes associated to ECM organization, chemotaxis or fibrosis in fibroblasts[35–39]. This second signature was assigned to IAFs. Spatial representation of the two signatures in the thyroid sections showed a significant increased presence of myofibroblast-like cells within the connective tissue, but also in the thyrocyte area, especially in GD samples (mean = $0.431 \pm 0.844$) compared to HT (mean = $-0.046 \pm 0.770$) and controls ((mean = $-0.035 \pm 0.725$), $p$ adjusted $< 0.001$ in both cases), which suggested their presence in the interfollicular space (Fig. 3e, gene signatures in Supplementary Data 2). On the other hand, IAFs were significantly overrepresented in the connective tissue of HT samples (mean = $0.051 \pm 0.429$) compared to GD (mean = $-0.023 \pm 0.362$) and controls ((mean = $-0.033 \pm 0.280$); $p < 0.001$ in both cases) (Fig. 3f). To validate the putative presence of myofibroblasts and IAFs and to rule out their association to other cells, we analyzed these signatures in HT single cell data and annotated them based on top markers and enrichment analysis into four cell types: SMCs, myofibroblasts, IAFs related to adhesion and chemoattraction, and IAFs related to ECM organization (Supplementary Fig. 11a, b). When we plotted our ST signatures from CT in HT single cell data, their expression matched with those cell subsets associated with SMCs/myofibroblasts and IAFs respectively (Supplementary Fig. 11c). We plotted main genes individually to further characterize these subpopulations: myofibroblasts were enriched in *ACTA2* and *ADIRF* with low levels of *MYH11*, and IAFs overexpressed *DCN*. There were two subpopulations of IAFs, one related to adhesion and chemotaxis with increased *CXCL12* levels, and a second subpopulation related to ECM reorganization with upregulated *IGFBP6* (Supplementary Fig. 11d). Deconvolution of HT samples located myofibroblasts in TFCs zones close to the immune infiltrate. *CXCL12* + IAFs seemed to distribute close to and even inside the immune infiltrate, whereas *IGFBP6* + IAFs were restricted to connective tissue regions (Fig. 3g).

To confirm the existence and the differential distribution across conditions of myofibroblast and IAFs subpopulations, we performed a multicolor immunofluorescence analysis. In control samples, CD34+ fibroblasts were present in the interstitial space surrounding thyroid follicles and did not express myofibroblast markers (α-SMA, TAGLN, and ADIRF). In contrast, in AITD CD34+ fibroblasts distributed in the interstitial space between thyroid follicles expressed high levels of α-SMA+ and TAGLN+ markers characteristic of myofibroblasts (mean α-SMA IHC score in HT $0.964 \pm 0.625$ and GD $1.352 \pm 1.060$ vs $0.266 \pm 0.487$ in control thyroid tissue, $p = 0.004$ and $0.001$, respectively) (Fig. 4a–c and Supplementary Fig. 12). Interestingly, a special subpopulation of myofibroblast that expressed ADIRF+ was significantly increased in GD tissue compared to both HT and controls (mean ADIRF IHC score in GD $1.552 \pm 0.797$ vs $0.593 \pm 0.647$ in HT thyroid tissue and $0.398 \pm 0.630$ in control thyroid tissue, $p < 0.001$ in both cases, Fig. 4a, b, d). Accordingly, a strong direct correlation between the IHC score of both ADIRF and a-SMA in GD samples was observed ($r = 0.8095$; $p < 0.0001$, Fig. 4e). Interestingly, clinical parameters such as TPOAB showed a direct correlation with both ADIRF and α-SMA scores ($r = 0.3243$; $p = 0.0297$ and $r = 0.4079$; $p = 0.0254$,

respectively. Figure 4e) Overall, these results confirmed that while generic myofibroblasts are increased in both HT and GD, specific ADIRF+ myofibroblasts are enriched in GD and are almost exclusively present in this condition.

We then validated the presence of IAFs, using DCN as a general marker of this cell subset[40]. In addition, we studied two different subpopulations, chemoattractant IAFs that co-expressed CXCL12 and ECM reorganizers IAFs that co-expressed IGFBP6. DCN was significantly increased in HT connective tissue compared to controls ($p = 0.0256$). No difference was observed between HT and GD ($p = 0.3147$) and GD vs controls ($p = 0.4615$) (Fig. 5a). Both CXCL12+ and IGFBP6+ IAFs were almost exclusively distributed in HT connective tissue compared to controls and GD (Fig. 5b, c). Interestingly, cell-cell communication analysis detected an increased number of cell traffic interactions between CT clusters (senders) and TILs (receivers) in HT and GD samples (Fig. 5d). In detail, one of the strongest interactions corresponded to the *CXCL12-CXCR4* pathway from CT fibroblasts to immune cells from other regions of the tissue, especially in HT (Fig. 5d, e and Supplementary Fig. 13).

Altogether, these results validated the transcriptomic data, where we confirmed the increased presence and established the precise spatial distribution of two fibroblasts subpopulations associated with inflammation in HT.

## Vascular molecular profiling displayed specific markers in AITD related to angiogenesis and increased permeability

Vessel-annotated spots were clustered into five different groups at 0.3 resolution (Fig. 6a). Notably, we did not find substantial differences between conditions (Fig. 6b). DE and pathway enrichment analysis among clusters showed that cluster V0 was enriched in mitochondrial activity-associated genes; V1 exhibited markers related to SMCs (*ACTA2, MYH11, TAGLN, ADIRF*), V2 and V3 probably presented mixed spots with TFCs, since top genes were related to thyrocytes (i.e., *TPO, TG, TFF3*), and with TILs (with genes such as *TRBC2, CD52* or *CD69*), respectively. V4 displayed gene sets related to angiogenesis and endothelial development (Supplementary Data 6).

To further identify differences within vessels between AITD and control tissues, we performed a DE analysis between conditions using vessel pseudobulks. We showed that genes related to the ETC complex III, such as *UQCR11, UQCRQ*, and *UQCRH*, which have been reported to be essential in angiogenesis[41], were upregulated in AITD compared to controls (Fig. 6c, d). Indeed, they were more highly expressed in HT (median = $0.459$ [$0.106, 0.736$]), compared to GD (median = $-0.001[-0.468, 0.517]$) and controls (median = $-0.266[-0.567, 0.040]$) ($p$ adjusted $< 0.001$; Fig. 6e). In addition, *ACKR1*, an endothelial marker related to High Endothelial Venules (HEVs) previously described in HT tissue[27], was upregulated in HT vessels compared to control samples, a condition almost exclusively associated with this disease (Fig. 6c). Moreover, EGF Like Domain Multiple 7 (*EGFL7*), a blood vessel inducer[42], and markers previously associated to angiogenesis and permeability in AITD such as endoglin (*ENG*)[25] were also upregulated in HT and GD vs controls (Fig. 6c, d). It is worth highlighting the significant expression in HT and GD vessels of Plasmalemma Vesicle Associated Protein (*PLVAP*) (Fig. 6c, d), an endothelial cell specific protein associated with diaphragmatic fenestrated vessels[43–45] that has not been previously reported in an AITD context.

General DE analysis between HT and GD confirmed the upregulation of *ACKR1* in HT; meanwhile *PLVAP* was enhanced in GD (Fig. 6f). Concerning spatial distribution in the slides, we observed that *ACKR1* was highly expressed in vessels and connective tissue areas of HT samples, as expected, whereas *ENG* and *EGFL7* had a wide expression in the TFCs area of AITD samples, (Supplementary Fig. 14). Interestingly, *PLVAP* also showed broad expression not only in vessels spots but also in TFCs and CT areas, with particularly high levels observed in GD samples (Fig. 6g). To validate these markers, we analyzed the different

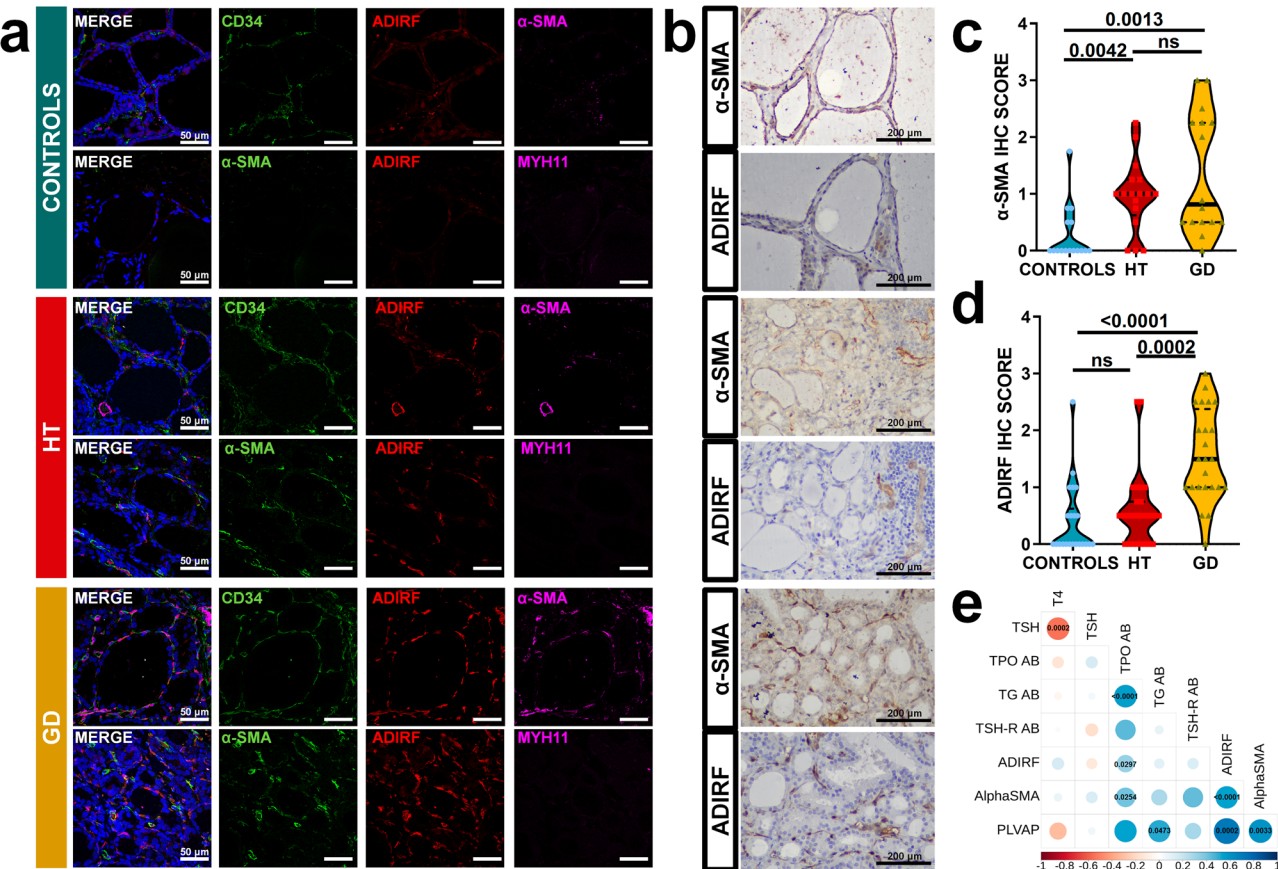

**Fig. 4 | Immunostaining of myofibroblasts markers from HT, GD, and control thyroid tissues. a** Representative confocal immunofluorescence images, in healthy control, HT and GD tissue samples, of the myofibroblast markers a-SMA (magenta), and ADIRF (red) in combination with a mesenchymal marker, CD34 (green) or a smooth muscle cell marker, MYH11 (magenta). Nuclei are stained with DAPI (blue). Objective: 63X. Scale bar: 50 μm. **b** Immunohistochemistry of ADIRF and a-SMA in serial tissue sections form controls, HT, and GD patients. Scale 200 μm. **c**, **d** Quantification of α-SMA and ADIRF in thyroid tissue from controls, HT, and GD patients. The quantification is described in Methods section. Kruskal-Wallis test and

Dunn´s test for statistical analysis and adjusted two-sided *p* values. **e** Analysis of correlation between clinical parameters and immunohistochemical scores with Spearman rho test and two-sided *p* values. The area of the circles shows the absolute value of corresponding correlation coefficients, Data are expressed as the arithmetic mean ± SD. Stainings were confirmed in at least seven biological replicates. HT Hashimoto's thyroiditis, GD Graves´ disease, TSH thyroid stimulating hormone, TPOAB thyroid peroxidase autoantibodies, TGAB thyroglobulin autoantibodies, TSH-R AB thyroid stimulating hormone receptor autoantibodies.

molecular profiles of the endothelial cells (ECs) from HT scRNA-seq data. We corroborated previous findings and identified three clusters: *ACKR1* + ECs, capillary *FLT1* + (Fms Related Receptor Tyrosine Kinase 1) and arterial *SEMA3G* + (Semaphorin 3 g) (Supplementary Fig. 15a, b). *EGFL7* and *ENG* were exclusively and highly expressed in capillaries and *ACKR1* + ECs (Supplementary Fig. 15c). Interestingly, *PLVAP* expression was upregulated mostly in capillaries (Supplementary Fig. 15c). This correlation might be attributed to fenestrated capillaries that increase their permeability. Moreover, the deconvolution analysis of the HT spatial samples revealed capillaries distributed in TFCs areas as suggested, whereas *ACKR1* HEVs distributed outlining the TILs regions (Supplementary Fig. 6c).

In agreement with ST data, immunostaining of thyroid sections showed a significantly higher number of PLVAP+ capillaries within the interstitial space surrounding thyroid follicles in AITD compared to control thyroid tissue (mean PLVAP+ capillaries in HT 7.438 ± 2.607 and GD9.714 ± 3.646 vs 4.429 ± 2.472 in controls; *p* = 0.003 and *p* = 0.002, respectively, Fig. 6h). Although we did not find significant differences between GD and HT, a trend of increased number of PLVAP + capillaries in GD was observed (mean PLVAP+ capillaries in HT 7.438 ± 2.607 vs 9.714 ± 3.646 in GD, *p* = 0.064) (Fig. 6h). Interestingly, no PLVAP expression was found in a-SMA+ arteries from AITD and control tissues, confirming our previous transcriptomic results

(Fig. 6i). Moreover, PLVAP score showed a strong direct correlation with both ADIRF and α-SMA scores (*r* = 0.7136; *p* = 0.0002 and *r* = 0.5972; *p* = 0.0033, respectively, Fig. 4e) and also with TGAB levels (*r* = 0.5192; *p* = 0.0473, Fig. 4e).

## Spatial distribution of thyroid infiltrating immune cells

To evaluate the spatial distribution of the immune cells, we considered that immune infiltration within the thyroid gland comprises two histologically differentiated zones: TILs areas (HT and GD samples) and GC forming tertiary lymphoid organs (HT samples). After that, we clustered both zones and deconvoluted their spots using single-cell signatures: TCD4, TCD8, NKs cells, *ACKR1* + HEVs, myeloid cells, germinal center B cells, proliferative B cells, naïve B cells, and plasmablasts.

To address the molecular differences between TILs in AITD and their potential implications, we conducted unsupervised clustering at a 0.3 resolution through sample integration (Supplementary Data 7). Intriguingly, we identified six clusters (Fig. 7a, b): I0, predominantly found in HT samples, histologically encompassed both mantle and infiltration areas. Indeed, I0 exhibited upregulation of genes more associated with FDCs and B cells (*FDCSP, CXCL13, VPREB3, CD19*), as well as infiltrating T cells (*TRAC, CD3D, CD3E, CCL21*), suggesting mixed spots (Fig. 7c, e). We aimed to molecularly separate the mantle zone

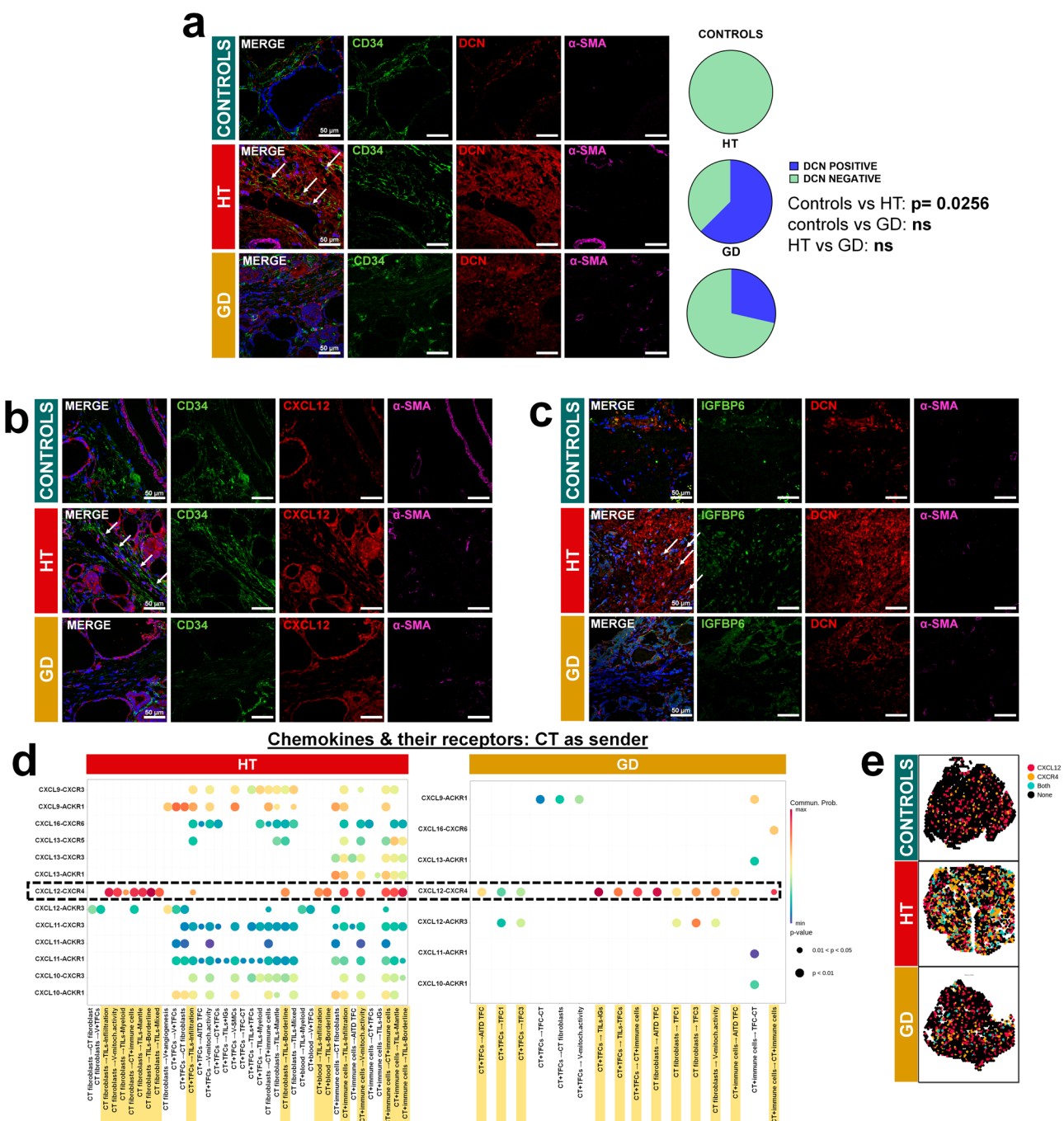

**Fig. 5 | Immunostaining of different IAF markers in HT, GD, and control thyroid tissues. a** Representative confocal immunofluorescence images, in healthy control, HT and GD tissue samples, of the IAFs marker DCN (red) combined with a mesenchymal/fibroblast marker, CD34 (green) and a myofibroblast marker, α-SMA (magenta). Nuclei are stained with DAPI (blue). Arrows indicate colocalization of CD34 and DCN. Objective: 63X. Scale bar: 50 μm. Right panels, quantification of DCN in thyroid tissue samples. Quantification is described in Methods section.$\chi^2$-square test for statistical analysis ($p = 0.0338$) Fisher´s exact test for individual comparisons and adjusted two-sided $p$ values. **b** Representative confocal immunofluorescence images, in healthy control, HT, and GD tissue samples, of CXCL12 (red) in combination with a mesenchymal/fibroblast marker, CD34 (green), and a myofibroblast marker, α-SMA (magenta). Nuclei are stained with DAPI (blue). Arrows indicate colocalization of CD34 and CXCL12. Objective: 63X. Scale bar: 50 μm. **c** Representative confocal

immunofluorescence images, in healthy control, HT, and GD tissue samples, of IGFBP6 (green) in combination with DCN (red) and a myofibroblast marker, α-SMA (magenta). Nuclei are stained with DAPI (blue). Arrows indicate colocalization of IGFBP6 and DCN. Objective: 63X. Scale bar: 50 μm. Stainings were confirmed in at least seven biological replicates. **d** Chemokines sent by CT clusters and their receptors in HT and GD samples. *CXCL12-CXCR4* is depicted with dashed lines, and significant interactions between regions are highlighted in yellow. Circle size corresponds to the $p$ value, and probability scores are indicated by color. **e** Spatial plot of *CXCL12* and *CXCR4* in controls, HT, and GD samples. AITD autoimmune thyroid diseases, HT Hashimoto´s thyroiditis, GD Graves´ disease, CT connective tissue, TFCs thyroid follicular cells, TILs thyroid infiltrating lymphocytes, V vessels, SMCs smooth muscle cells, IGs immunoglobulins.

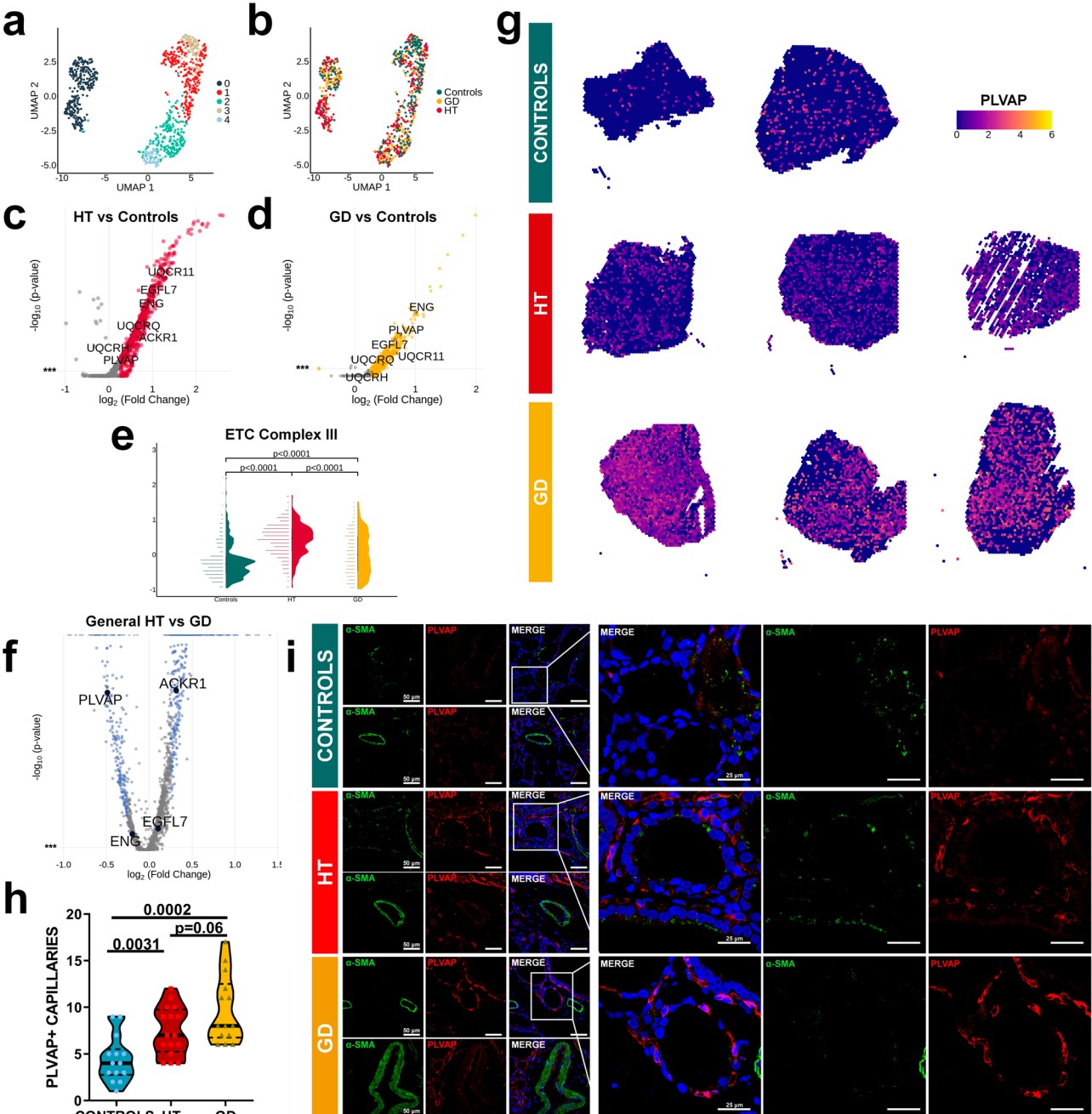

**Fig. 6 | Analysis of ST data from vessel annotated-spots and distribution of PLVAP+ capillaries. a** Clustering at 0.3 resolution of the vessels zones from spatial transcriptomic data of AITD and control samples. **b** Distribution of conditions across the clustering of vessel zones. Volcano plots and remarkable genes of the differential gene expression within vessels areas between (**c**) HT and controls and (**d**) GD and controls. **e** Representation of the expression signature of ETC complex III genes (*UQCR11, UQCRQ, UQCRH* genes) across conditions within vessels areas. Dunn test: HT (*n* = 3): spots = 404, median = 0.459 [0.106, 0.736]; GD (*n* = 3): spots = 211, median = −0.001 [−0.468, 0.517]; controls (*n* = 2): spots = 254, median = −0.266[−0.567, 0.040]; all *p* adjusted by Holm test <0.001. **f** Volcano plot of the general differential expression analysis between HT and GD. **g** Spatial

distribution of *PLVAP* in control, HT and GD samples (**h**) Violin plots comparing the quantification of PLVAP+ capillaries between the three groups. T test with Welch´s correction for statistical analysis and adjusted two-sided *p* values (**i**) Representative confocal immunofluorescence images, in healthy control, HT, and GD tissue samples, of PLVAP (red) and α-SMA (green). Nuclei are stained with DAPI (blue). Objective: 63X. Scale bar: 50 μm, zoom: 25 μm. Stainings were confirmed in at least seven biological replicates. Adjusted *p* value with Bonferroni (**c, d, f**) ST spatial transcriptomics, HT Hashimoto´s thyroiditis, GD Graves' disease, UMAP Uniform Manifold Approximation and Projection for Dimension Reduction, ETC electron transport chain.

from the immune infiltrate in cluster I0 by performing a re-clustering at a 0.1 resolution. I0 was divided into a cluster closer to the GC, which upregulated FDC-associated genes (Fig. 7d, cluster 1), and another distributed over the infiltrate enriched in *CCL19, FOS,* and *CCL21,* among others (Fig. 7d, cluster 0 and Supplementary Data 8). Clusters I1 and I2 comprised both HT and GD spots and occupied borderline

zones between dense immune infiltrated areas and gland stroma, characterized by immunoglobulin genes. I2 exhibited high content of thyrocyte markers (i.e., *TPO, PAX8, TG*). Cluster I3, represented by HT spots, was located in the borderline areas between the immune infiltrate and gland stroma, and showed upregulation of T cell-associated genes (*TRAC, TRBC1, CD3D*) and B cell markers (*MS4A1, CR2*). Finally,

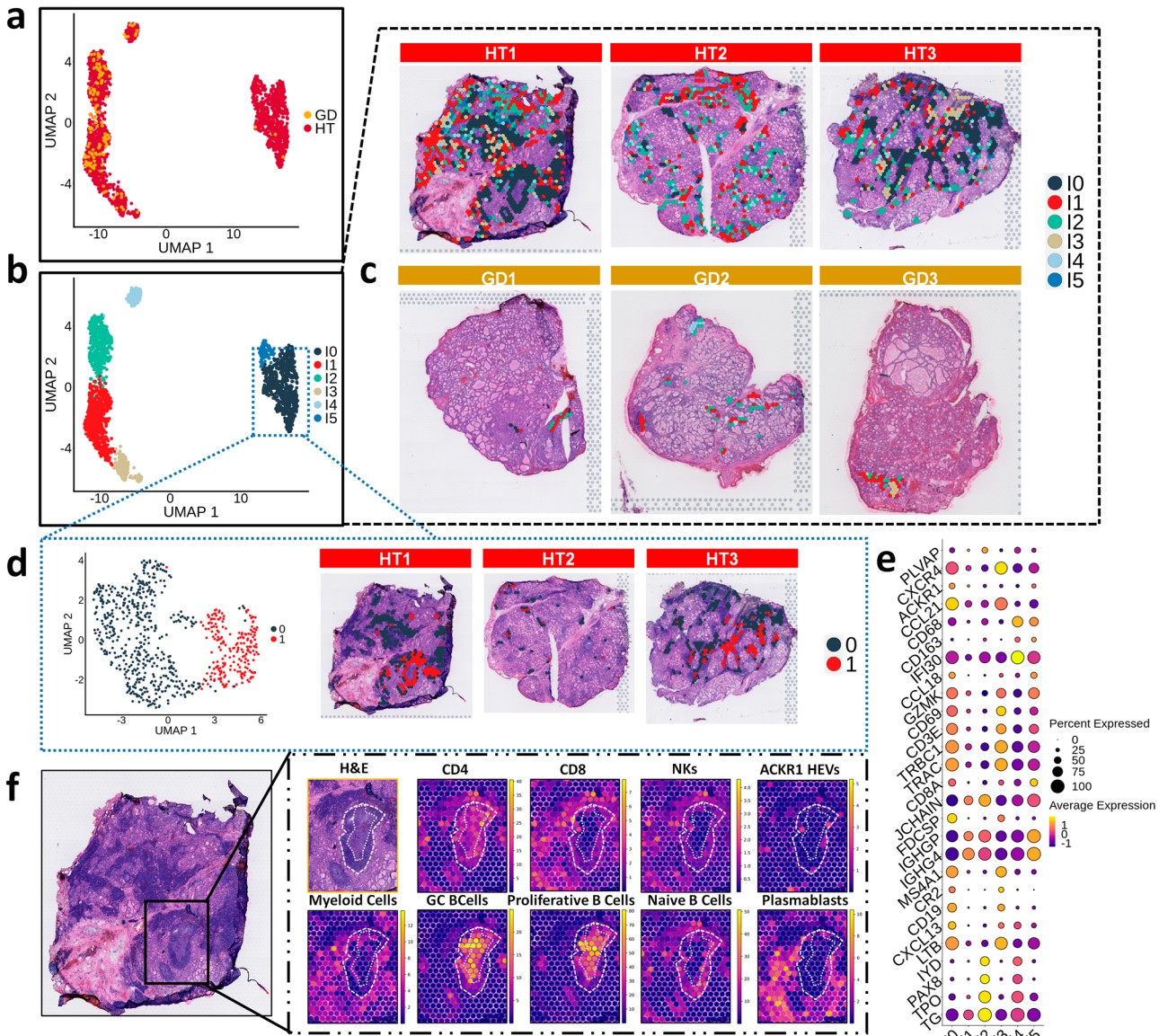

**Fig. 7 | Analysis of TILs and GC in HT and GD samples. Deconvolution of the spots using immune cells signatures. a** UMAP of all the spots of the TILs regions from HT and GD samples. **b** UMAP colored by the different clusters at 0.3 resolution. **c** Spatial mapping of TILs clusters. **d** UMAP and spatial mapping of I0 reclustering at a resolution of 0.1. **e** Dot plot of representative markers of immune and stroma cells across clusters of TILs. **f** Hematoxylin and eosin and GC region detail showing the proportions of deconvoluted immune cells using signatures from scRNAseq analysis. Dotted white lines delineate the mantle (between outer and inner lines) and the GC, including the light and dark zones (inner line). H&E hematoxylin & eosin, HT Hashimoto´s thyroiditis, GD Graves´ disease, UMAP Uniform Manifold Approximation and Projection for Dimension Reduction, GC Germinal Center, TILs thyroid infiltrating lymphocytes, HEVs high endothelial venules.

clusters I4 and I5 incorporated a low quantity of spots: I4 showed upregulation of myeloid markers (*CXCL10*, *SPP1*, *IFI30*) and I5, exclusive of HT samples, increased expression of immunoglobulin genes, myeloid (*CD68*) and T cell markers (*CD8B*, *GZMA*) (Fig. 7e). DE analysis of pseudobulk of TILs regions between HT and GD did not provide meaningful results (Supplementary Data 9).

In GC areas, the most upregulated markers corresponded to FDCs (Supplementary Data 1). Spatial location of the top five GC markers revealed a specific concentration of these genes in GC regions, especially *FDCSP* and *TCL1A*, suggesting a precise location of FDCs and B cells (Supplementary Fig. 16).

To shed light into the distribution of immune cells, we performed spot deconvolution in HT samples. Focusing on the histological areas, GC was enriched in proliferative B cells, GC B cells, and TCD4 (Fig. 7f and Supplementary Fig. 6a). The canonical FDC markers were not

detected in the scRNA-seq data, therefore we were not able to map the spatial localization of these cells in the deconvolution analysis. The mantle zone surrounding the GC exhibited an enhanced distribution of naive B cells, TCD4 and, to a lesser extent, TCD8 and NK cells (Fig. 7f). Other dense immune zones also exhibited a higher proportion of T cells (Fig. 7f). Remarkably, myeloid cells (DC and macrophages) were distributed surrounding the infiltrate, whereas *ACKR1* + HEVs were located within TILs regions and CT areas. Additionally, plasmablasts disseminated within TFCs areas (Fig. 7f and Supplementary Fig. 6a).

Regarding the unavailability of scRNAseq data from the thyroid gland of GD patients to perform spot deconvolution, we plotted different immune gene markers in our ST of GD samples to elucidate their location. Remarkably, our results showed abundant plasma cells not only in infiltrated areas, but also in TFCs regions close to the immune infiltrate and connective tissue. Moreover, we detected a significant

expression of myeloid cell markers across the tissue in these samples (Supplementary Fig. 17).

## Discussion

In the present study, we have established a transversal approach to understand the cellular specificity and complexity of thyroid tissue microenvironment to increase our knowledge of the pathogenic mechanisms of AITD.

The heterogeneity of cells in a given tissue poses a major challenge to understand the pathogenesis, make an accurate diagnosis and establish personalized treatment strategies. Thyroid tissue in AITD includes different cell populations such as TFCs, various types of immune cells, and stromal cells, including fibroblasts and endothelial cells that are highly interdependent compartments. The identification of abnormalities in gene expression or in cellular interactions in the thyroid tissue in these patients may not always be reflected in changes in systemic serum markers such as thyroid hormone levels or the presence of thyroid auto-antibodies. These tissue-specific abnormalities, may help clinicians to better understand why some patients continue to experience symptoms despite seemingly adequate treatment. While bulk analyses of thyroid tissues have provided important insight into the pathogenic mechanisms involved in AITD, they do not differentiate between pathogenic, damaged, and normal cells, nor can they distinguish between different cell populations in mixed tissues. Recent scRNA-seq reports in HT and GD[27,46], have improved our knowledge on the contribution of TFCs and stromal cells to shape the autoimmune response. In our work, we study the molecular interplay between known cell types present in AITD thyroid tissue at spatial resolution. Here, we not only aimed to provide critical insights into thyroid biology per se, but also to shed light on other therapeutically relevant issues related to heterogeneity such as the interplay between TFCs, connective tissue, vascular and immune cells.

In the present study, the evaluation of a TDS signature in AITD revealed a subset of de-differentiated thyrocytes in HT epithelium. These damaged TFCs presented an increased expression of genes related to oxidative stress responses, such as glutathione selenoproteins (i.e., *GPX4*) and respiratory chain activity genes (i.e., ETC complex I and IV genes *NDUFB2*, *COX8A* or *COX5B*). Indeed, previous studies reported that some of these alterations were associated to oncocytic regions in HT samples[47]. Intriguingly, this subpopulation of TFC was spatially close to the immune infiltrated areas of the tissue. In this regard, a single-cell analysis in papillary thyroid carcinoma (PTC) has recently reported the existence of premalignant thyrocytes (described as cells with low TDS, enriched in pathways associated with stress response and upregulated *TMSB4X*), without morphological or histological changes before cancer formation[30]. Moreover, other studies reported an increase in CD74 in TFCs associated to malignancy[48] and resistance to apoptosis by mitochondria stabilization[49]. Overall, the possible existence of a subpopulation of premalignant thyrocytes is in line with our observations of damaged/poorly differentiated TFCs, suggesting a possible link of the association of GD and HT with PTC, as previously reported[50,51].

Interestingly, TFCs from patients with AITD presented an increase in the expression of CD74 both by ST and by IHC/IF. CD74 is expressed by several cell types including immune, mesenchymal, endothelial, or epithelial cells and they have been involved in several processes. In addition, CD74 has been involved in antigen presentation, associated with the Major Histocompatibility Class II antigen complex (MHC-II)[52], widely expressed in TFCs of AITD patients[53,54]. Indeed, TFCs in HT exhibited interactions related to antigen presentation (*HLA-CD4* immune cells). Thus, CD74 overexpression in TFCs could be related to antigen presentation pathways in HT, whereas in GD it may be involved in the presentation of self-antigens via HLA without costimulatory signaling resulting in tolerance rather than activation, as recently described[46].

The microenvironment associated with AITD might contribute to the upregulation of the CD74/MIF pathway. Proinflammatory cytokines such as IFN-γ and TNF-α can upregulate the expression of CD74 in TFCs and also in immune cells infiltrated in the gland[55,56], which enhances the interaction of this receptor with its ligand, MIF. Their interaction can also help in tissue repair in different auto-immune diseases such as Inflammatory Bowel Diseases (IBD) or type I diabetes (T1D)[57,58]. The results obtained from the cell communication analysis revealed that AITD-associated TFCs from HT samples were MIF senders but not receivers. On the other hand, AITD-associated TFCs in GD samples were both MIF senders and receivers. Thus, the interaction CD74/MIF in AITD could contribute to opposed processes: exacerbation of thyroid infiltration by immune cells (especially in HT) and promotion of autocrine TFC repair in GD, as demonstrated in IBD intestinal mucosa and in skin epithelium wound healing[57,59]. Accordingly, CD74/MIF may act on autoimmune tissue damage as a double-edge sword priming its persistence but also favoring TFCs regeneration. The identification of these targets may represent an opportunity for the development of innovative therapies aimed to disrupt antigen presentation by TFCs to thyroid TILs, including gene therapies, small molecule inhibitors, or biologics such as antibodies against CD74[48,60,61].

Fibroblasts are key players in inflamed microenvironments[62–65]. An important finding in our analysis is the identification of two main fibroblasts subpopulations in connective tissue: ADIRF+ myofibroblasts surrounding thyroid follicles and IAFs within connective tissue regions. Myofibroblasts have physiological relevant functions, as they are related to wound healing or regulation of ECM in combination with the development of new vessels for tissue repair in the epithelium[66,67]. In pathological conditions, myofibroblasts are persistently activated and accumulated in the affected tissue, spreading to unaffected areas and producing an excessive deposition of ECM, which leads to the generation of a fibrotic area[68]. Actually, previous works have demonstrated their presence in autoimmune diseases, such as rheumatoid arthritis[69,70], fibrotic diseases (i.e., cirrhosis)[67,71] and cancer (where they are also known as cancer myofibroblasts or myoCAFs)[72]. Myofibroblasts expressing a-SMA and Collagen I have been previously described to be variably represented in AITD samples. In this regard, TGF-β has also a role in myofibroblast generation, as this cytokine enhances the transition of thyroid fibroblasts to myofibroblasts[73]. Indeed, studies in autoimmune thyroiditis mouse models have described the contribution of myofibroblasts in tissue fibrosis expansion[74]. Recently, Zhang et al. described a subset of stromal cells in HT tissue, composed of myofibroblasts, partly surrounding ACKR1+ HEVs, which could contribute to the formation of tertiary lymphoid organs[27].

Our results not only report a significant increase of myofibroblasts in AITD in the interfollicular space, but also the presence of ADIRF+ myofibroblasts as a major cell subpopulation in GD patients. This protein is commonly related to adipose tissue as it stimulates the expression of the adipogenesis master regulators Peroxisome Proliferator-Activated Receptor Gamma (*PPARG*) and CCAAT/enhancer binding protein alpha (*CEBPA*). ADIRF is associated with myofibroblast-like cells in several tissues, such as pancreas and bladder[39,75–77]. In pancreas, ADIRF expression has been attributed to quiescent pancreatic stellate cells (PSCs) expressing adipogenic genes[75]. PSCs drive the severe fibrous reaction of Pancreatic Ductal Adenocarcinoma (PDAC)[78] and ADIRF+ PSCs have been recently described to be increased in advanced stages of PDAC[77]. One plausible hypothesis for the increase in ADIRF+ myofibroblasts in GD is that these cells may remain in an inactivated state, making them more predisposed than quiescent fibroblasts to be reactivated by an injury event. In this regard, studies in liver fibrosis have described that inactivated hepatic stellate cells upregulate adipogenesis markers such as PPARγ[79], which has been previously related to ADIRF mediated activation[80]. Thus, ADIRF+ myofibroblasts in GD

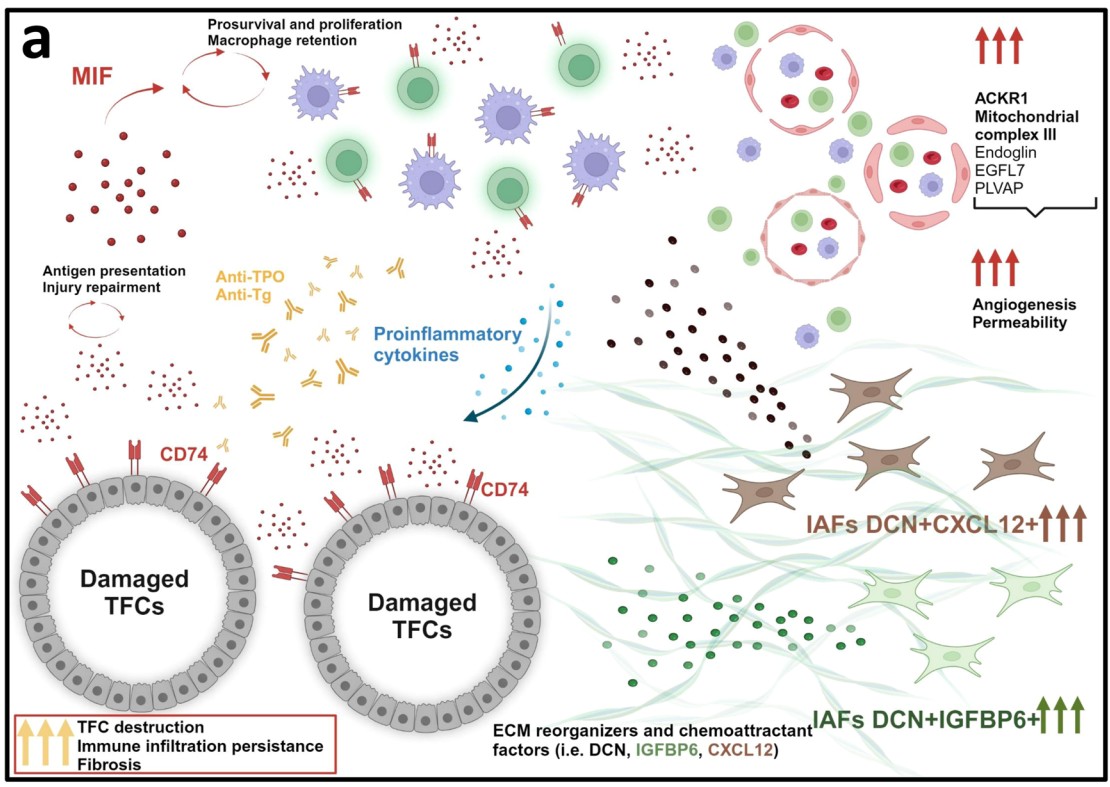

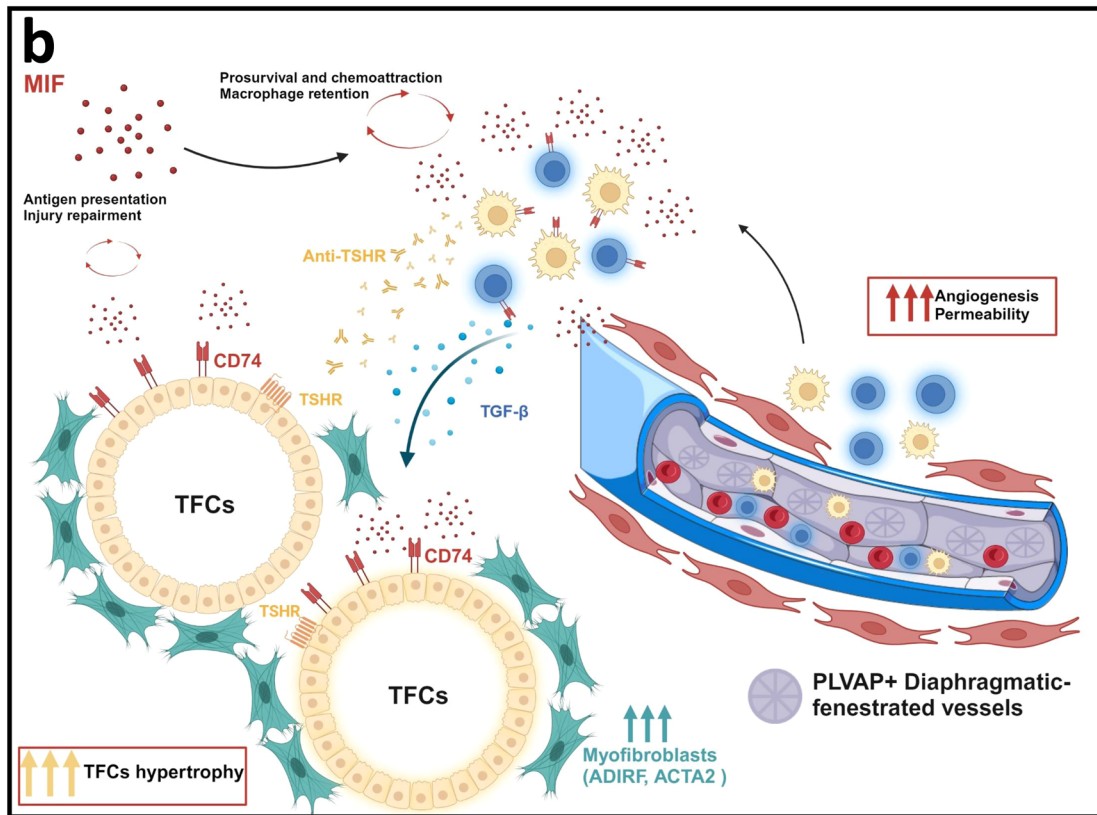

may represent an inactivated state of myofibroblasts that could be more easily reactivated in a pro-inflammatory scenario.

The most represented fibroblast subpopulations in HT were IAFs. IAFs are tightly related to a subtype of fibroblasts called inflammatory cancer-associated fibroblasts (iCAFs), key elements in the pathogenesis of some carcinomas[81]. In this context, iCAFs are able to secrete chemokines, such as CXCL12, associated with chemotaxis and immune infiltration in the tumor microenvironment. However, this immune infiltration is not associated with a better prognosis, probably due to the immunosuppressive environment[39]. Furthermore, in papillary thyroid carcinoma, a group of iCAFs with a remarkable expression of *CFD*, *DCN*, or *CCDC80*, facilitated the migration of tumor cells through ECM degradation[82]. Moreover, fibroblasts associated with inflammation have been reported in a cross-tissue single-cell atlas that

**Fig. 8 | Schematic hypothetical model of the role of different cell compartments in AITD pathogenesis.** HT model (**a**) and GD model (**b**). After immune tolerance breakdown and damage of thyroid tissue, TFCs may induce the expression of CD74 that could interact with its ligand MIF through an autocrine loop. Their interaction contributes to tissue repair but also can promote chemoattraction of additional immune cells to the gland. This positive feedback might enhance the immune response against the thyroid, via MHC mediated antigen presentation. In GD, the proinflammatory environment would promote the proliferation of fibroblasts and their differentiation into myofibroblasts (ADIRF- and ADIRF+). Alternatively, in the case of HT, fibroblasts will differentiate mostly to IAFs that would synergize with immune cells in TFCs destruction and also will contribute to the generation of fibrotic tissue, and to the production of chemoattractant and proinflammatory cytokines. In both diseases, angiogenesis would be enhanced accompanied by an increased vessel permeability with some differences between them: PLVAP+ fenestrated vessels appear mainly in GD while ACKR1+ HEVs are mainly present in HT. HT Hashimoto´s thyroiditis, GD Graves´ disease, TFCs thyroid follicular cells, TGF-β transforming growth factor β. Created with BioRender.com, released under a Creative Commons Attribution-NonCommercial-NoDerivs 4.0 International license.

encompassed patients with autoimmune disease such as rheumatoid arthritis, Sjögren's syndrome, and ulcerative colitis[83]. We defined two IAFs (chemoattractant CXCL12+ and ECM reorganizers IGFBP6+) that were remarkably increased close to immune infiltration and within connective tissue compartments in HT patients, respectively. In particular, we identified an increase of IAFs expressing CXCL12 that may be related to chemotaxis processes. Indeed, this cytokine is able to recruit and promote the accumulation of leukocytes via CXCR4/CXCR7[84] and participates in the organization of ectopic secondary lymphoid follicles that takes place in autoimmune diseases, such as in AITD[85,86]. The results of our ligand-receptor analysis confirmed the interaction between fibroblasts and immune cells, as connective tissue fibroblasts were *CXCL12* senders and immune cells appeared as *CXCL12* receivers via *CXCR4*. Concerning IGFBP6+ IAFs, this subpopulation is associated with ECM remodeling, in particular with ECM destruction and pro-fibrotic properties[35]. Moreover, this subtype might share immune cell chemoattractant properties with CXCL12+ IAFs, as IGFBP6 is able to induce a greater in vitro migration of T lymphocytes derived from RA patients[87]. Overall, as fibrosis and the presence of immune infiltrated cells within tissue are typically observed in HT pathogenesis[88], we hypothesize that HT-associated IAFs may be tightly interconnected with both pathological processes.

As TFCs and connective tissue, vessels, and angiogenesis play an essential role in AITD pathogenesis[89]. In fact, angiogenesis is a well-known process in hypertrophic organs, as occurs in GD where thyroid growth needs the development of new vessels in order to support organ expansion[90]. In line with our observations, previous works have highlighted the relevance of the angiogenic process in both AITD, with greater significance observed in GD patients[91]. Furthermore, angiogenesis and myofibroblasts are tightly interconnected in physiological and pathological processes such as wound healing[66,92]. We also found an angiogenesis-related signature within TFCs in our ST analysis with a significant increase of *PLVAP* in the interfollicular capillaries of AITD tissue. PLVAP is a permeability-associated molecule in diaphragmatic-fenestrated vessels that regulates transendothelial traffic of solutes and immune cells, as well as angiogenesis through endothelial cells[44,45,93,94]. We identified an increase of PLVAP+ capillaries in AITD, especially in GD samples. Interestingly, PLVAP+ endothelial cells have been related to fibrotic processes and can enhance leukocyte transmigration in liver cirrhosis[95]. Besides, an in silico analysis of PTC with and without concurrent HT described an increase of *PLVAP* in vascular endothelial cells in both conditions, related to an induction of lymphocyte infiltration[50].

TILs have been deeply studied in AITD[14–20]. In the present study, we were able to map the major immune cell types into the spatial transcriptome in accordance with other works based on immunostaining[6–9]. Spot deconvolution helped us to unfold the molecular complexity of the immune infiltrate and tertiary lymphoid organs in AITD. Remarkably, we observed that these immune cells displayed interactions with AITD-associated TFCs (through *MIF* and *MHC-II*) and stromal cells such as *CXCL12* + IAFs capable of promoting *CXCR4*+ cell recruitment.

One of the main limitations of our study is the low number of samples analyzed by ST. Furthermore, the technology used in our study does not provide single-cell resolution requiring spot deconvolution and prediction of cell type location by integrating single cell data to our ST analyses. To overcome such limitations, we have incorporated a larger cohort of samples of thyroid TMAs to ensure robust validation of ST data by IHC/IF. Additional studies are needed to elucidate the role of the different thyroid subpopulations found in AITD samples. Furthermore, further experimental evidence is needed to clarify the role of different cell populations in perpetuating the autoimmune disease.

In summary, our data contributes to our knowledge of AITD pathogenesis at both cellular and molecular levels under spatial resolution. We provide a dataset at spatial resolution that could serve as a resource to perform further studies in AITD patients. Our characterization of the different cell populations involved in both the HT and GD thyroid ecosystems (summarized in Fig. 8) could support the development of biomarkers and treatment strategies. Future functional analysis will be essential in order to discern the role of the different subpopulations described here.

## Methods

### Ethics approval and consent to participate
This project was approved by the Internal Ethics Review Committee of the Hospital de La Princesa (reference number: 4783) and written informed consent was obtained from all participants prior to inclusion, in accordance with the Declaration of Helsinki.

### Study participants
We used thyroid tissue samples from AITD lesions and histologically normal parenchyma adjacent to other thyroid lesions from patients who underwent thyroidectomy at the Surgical Department of the Hospital Universitario la Princesa (registry number C0003922). All thyroid tissues were reviewed by an experienced pathologist and diagnosis was confirmed by an endocrinologist using commonly accepted clinical, laboratory, and histological criteria. Serum free thyroxine 4 (FT4) and TSH levels were measured using Elecsys® FT4 II kit Cat# 06437281 190 and Elecsys® TSH Kit Roche Cat# 11731459 122, with a reference range of 0.93–1.7 ng/dl and 0.27–4.20 mU/ml, respectively. Titers of thyrotropin receptor (TSH-R), thyroglobulin (TG), and thyroperoxidase (TPO) antibodies were assessed by ELISA (Medizym TRA human Ref: #3505 from Medipan; EliA anti-TG Well Ref: #14–5642 and EliA anti-TPO Well. Ref: #14–5641, both from Thermo-Fisher Scientific, respectively).

This study was reviewed and approved by the Internal Ethics Review Committee of the Hospital de la Princesa (Committee Register Number: 2796) and informed consent was obtained from all participants in accordance with the Declaration of Helsinki.

### Sample preparation and sequencing
Thyroid tissue samples were immediately snap-frozen in liquid nitrogen cooled in isopentane and transferred to a −80 °C freezer. Eight samples were selected for ST analysis using the Visium spatial Gene expression Kits from 10X Genomics. RNA was isolated using miRNeasy Mini Kit (Qiagen Hilden, Germany). RNA quality and quantity were assessed in a 2100 Bioanalyzer using an RNA 6000 Nano kit (Agilent

Technologies. Santa Clara, CA, USA) to ensure a minimum RNA Integrity Number (RIN) of seven. Then, samples were cryosectioned at 10 µm onto 6.5 × 6.5 mm capture areas of Visium spatial slides and processed according to the manufacturer's instructions (10x Genomics. Pleasanton, California, USA). Slides were H&E stained before the sections were imaged using the NanoZoomer S60 (Hamamatsu. Shizuoka Pref. Japan) to assess tissue morphology and quality. Sections were then permeabilized for 3 min, according to the results of the Visium Spatial Tissue Optimization Slide & Reagent kit (10X Genomics, CG000238), and processed according to the Visium Spatial Gene Expression user guide (10X Genomics, CG000239).

Reverse transcription and second strand synthesis were performed on slides, followed by cDNA amplification and quality control, and the pooled libraries were sequenced on Nova-seq 6000 (Illumina) with a sequencing depth of 100,000 reads per spot.

The spatial transcriptome data from human thyroid tissues were deposited in the Gene Expression Omnibus Database with accession number GSE248205.

## Data preprocessing

High-quality images and raw ST data were processed using SpaceRanger 10x Genomics software version 1.2.2 and mapped to the human genome reference GRCh38. Quality control was evaluated using R (version 4.0.3) and Seurat package[96]. To define the sequencing quality scores of each sample we established the recommended threshold and quality control metrics from 10x Genomics. Then, we evaluated the spatial plotting of the number of counts, number of features, and percentage of ribosomal and mitochondrial genes. Low-quality spots were removed.

(Code available in https://github.com/endonutriHUPR/AITD_SpatialTranscriptomics.git).

## Histology-based region selection

Hematoxylin and eosin (H&E) thyroid images (Supplementary Fig. 1) were evaluated and annotated manually by two independently trained pathologists using Loupe browser software 5.0 interface (10x Genomics). Spots included in destructed or broken tissue, low-quality control metrics or tissue-fold artifacts were excluded from any further analysis (Supplementary Fig. 2). Samples were annotated for specific cell tissue populations according to their histology and then validated by their molecular/transcriptomic profile.

## Normalization, dimensional reduction, integration, and batch effect correction

Data logarithmic normalization (2000 variable features) of the eight samples was performed using *NormalizeData* from Seurat package (version 4.2.0)[96] in R (version 4.0.3). 3000 variable features were chosen to merge and scale all samples together. To correct batch effects of "samples" and "experimental day", we performed a Principal Component Analysis (PCA) with the top 30 dimensions and a Harmony integration[28].

## Clustering and differential expression within cell populations

To investigate specific cell states and map the heterogeneity of each compartment (TFCs, connective tissue, vessels and immune infiltration), we addressed the analysis as follows: (1) from the Harmony integration (previously assessing the number of dimensions by the elbow method), we applied *FindClusters* function at 0.1, 0.3 and 0.5 resolutions to identify main clusters followed by Uniform Manifold Approximation and Projection (UMAP) generation and re-clustering when required. Then, we identified cluster-defined genes by *FindAllMarkers* function with default options. (2) A pseudobulk DE analysis of each compartment through Wilcoxon algorithm between conditions. (3) A general DE considering the whole tissue between conditions. This step was also implemented in Seurat package (version 4.2.0)[96], in R (version 4.0.3).

## Biological enrichment analysis

We used clusterProfiler package[97] for gene set enrichment analysis (GSEA)[98] to identify gene sets and the different pathways disturbed in each tissue cell population. In addition, we explored specific pathways with KEGG[99] and Reactome[100] databases. *AddModuleScore* from Seurat was used to identify, quantify, and plot remarkable signatures. Bioinformatic analysis code is available at GitHub: https://github.com/endonutriHUPR/AITD_SpatialTranscriptomics.git.

## Single-cell analysis

Published single-cell data from the thyroid of 4 HT patients[27], accession number HRA001684 for the Genome Sequence Archive (GSA), were analyzed to validate cell populations obtained by ST results. DoubletFinder R package[101] was applied in each sample to filter doublets. Then, we followed the authors' guidelines using Seurat (version 4.2.0)[96] workflow and Harmony[28] to correct the samples' batch effects. Cell re-clustering was performed by *SplitObject*, followed by a logarithmic scale normalization and keeping the 2000 variable features. The samples were then merged again and scaled to perform a PCA and a new integration[28]. To identify and annotate the different clusters, we ran *FindAllMarkers*. Clusters identified as doublets were removed. GSEA[98], KEGG[99], and Reactome[100] were used for functional enrichment analysis with the package clusterProfiler[97].

## Cell-cell interactions and spot deconvolution

The CellChat database and R package[33] with default parameters were used to separately identify biological interactions between annotated clusters in each condition. We focused on the relationships of TFCs and CT regions with the rest of areas by *sources.use* and *target.use* arguments.

Spots deconvolution of the three HT samples was performed using cell2location[102] implemented in Python3. We used the annotation of the major cell types derived from our analysis of the sc-RNASeq HT published data[27] (Supplementary Fig. 3). As arguments, we specified 30 cells per location, a detection_alpha of 200, and 20000 epochs when mapping the cell signatures. Downstream plots depict the *q05_cell_abundance_w_sf*. Code is available at GitHub: https://github.com/endonutriHUPR/AITD_SpatialTranscriptomics.git.

## Tissue microarrays, immunohistochemistry (IHC) and immunofluorescence (IF) staining

To provide a high-throughput tissue analysis, tissue microarrays (TMA) were constructed with a total of 76 thyroid tissues (29 HT, 24 GD, and 23 controls) (clinical data in Supplementary Table 1). TMA serial sections were placed in an oven at 65 °C for 1 h, deparaffinized in xylene and rehydrated through graded alcohols. Antigen retrieval was performed in a PTLink instrument using EnVision Flex target retrieval solution high or low pH (Agilent. Santa Clara, CA, USA) according to the manufacturer's instructions. Thereafter, tissue cross-sections were blocked with 5% bovine serum albumin and 10% normal goat serum in PBS or 5% peroxidase diluted in methanol for 15 min at room temperature and incubated with the primary antibodies (dilutions, catalog numbers, and suppliers of the antibodies are summarized in Supplementary Table 2). The following day, slides were washed three times with PBS and incubated 1 h with Alexa Fluor or HRP-conjugated antibodies. For IHC, sections were incubated with the proper horseradish peroxidase-conjugated secondary antibodies, incubated with 3,3'-diaminobenzidine (DAB; Agilent. Santa Clara, CA, USA), and counterstained with hematoxylin (Sigma Aldrich. Saint Louis, MO, USA). Sections were observed using a Nikon Eclipse Ci microscope (Nikon. Tokyo, Japan). For IF, DAPI was used for nuclei counterstaining. 5–7 mm thick stacks from a Leica Sp5 confocal microscope were analyzed for marker quantification. Markers in different thyroid sections were estimated manually by analyzing Z-stacked images captured in the confocal microscope. Low and higher magnification images were

analyzed by two independent observers in a blinded manner. IHQ and multicolor immunofluorescence stainings were analyzed and confirmed in at least seven biological replicates.

Quantification was performed using the approximate percentage of positive cells and staining intensity. All the tissue areas from each TMA slide underwent observer-blind examination. The proportion of positively stained cells was scored as follows: for quantitative analyses (a-SMA and ADIRF) 0: <5% stained cells; 1: 5–33% stained cells; 2: 33–66% stained cells and 3: >66% stained cells. PLVAP quantification was performed by the number of capillaries positively or negatively stained. DCN expression was quantified by the frequency of positive and negative stained tissues.

### Statistical analysis

Descriptive results are expressed as mean ± standard deviation (SD) or median and 25th–75th percentiles [Q1, Q3], as appropriate. Spearman's bivariate correlations were performed for all quantitative variables and differences between groups were compared using analysis of variance. Normality was assessed using Shapiro-Wilk test. Student's $t$ test and ANOVA were used when distributions passed normality. Mann Whitney U – Wilcoxon, Dunn test, and Kruskal-Wallis were used in non-normal distributed groups. Holm test was used in adjusted $p$ values. Contingency analyses were performed for qualitative variables, and the differences between groups were compared using $\chi^2$ and Fisher's exact tests. Samples from all groups within an experiment were processed at the same time. The $p$ values are two-sided and statistical significance was considered when $p < 0.05$. All statistical analyses were performed using R version 4.0.3 and GraphPad Prism 9 (GraphPad Software, San Diego, California USA).

### Reporting summary

Further information on research design is available in the Nature Portfolio Reporting Summary linked to this article.

## Data availability

The Spatial transcriptomics sequencing dataset generated in this study have been deposited in the Gene Expression Omnibus (GEO) Database under accession code GSE248205. All other data included in this study are provided in Supplementary data and Source data. The scRNAseq repository used for validation[27] was deposited by their authors in the Genome Sequence Archive (GSA) Database under accession code HRA001684. Source data are provided with this paper.

## Code availability

All the codes are available at GitHub: https://github.com/endonutriHUPR/AITD_SpatialTranscriptomics.git and linked to Zenodo[103].

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

## Acknowledgements

We thank Francisca Molina-Jimenez from the Microscopy Unit of the Instituto Investigación de la Princesa (IISP) for technical assistance with confocal microscopy, and Manuel Gómez for English corrections. We also thank Alejandra Rosell from HUP Pathology Department for acquisition of samples, Marc Elosua Bayes for quality control of spatial transcriptomic data, and CNIC bioinformatic unit, especially to Carlos Torroja for single-cell analysis support. We also warmly thank all the participants included in the study for their selfless participation. This work was supported by the following grants: Proyectos de Investigacion en Salud PI19/00584, PI22/01404, and PMP22/00021 (funded by Instituto de Salud Carlos III), iTIRONET- P2022/BMD7379 (funded by Comunidad de Madrid), Research project IPI/2022/N5 (funded by Sociedad de Endocrinología, Nutrición y Diabetes de la Comunidad de Madrid-SEN-DIMAD) and co-financed by FEDER funds to M.M. and R.M.H., as well as Contratos Predoctorales de Formación en Investigación en Salud (FI20/00035 and FI23/00052) grants to P.S.G. and N.S.B. The funders had no role in study design, data collection, data analysis, interpretation or writing of the report.

## Author contributions

R.M.H., P.S.G., and A.S.S. conceived and carried out experiments. N.S.B., P.S.G., M.M., and R.M.H. designed and interpreted the data. N.S.B., F.S.C., and H.H. contributed to single-cell and spatial transcriptomics workflow. N.S.B. developed necessary computational and visualization tools. P.S.G., A.S.S., M.S.N., J.L.M.D.N., M.M., and R.M.H. collected samples and contributed with clinical and pathology expertise. N.S.B., P.S.G., M.M., and R.M.H. were involved in writing the paper. All authors critically reviewed the manuscript for important intellectual content and approved the final manuscript.

## Competing interests

H.H. is a co-founder and shareholder of Omniscope, a scientific advisory board member of MiRXES and Nanostring, and a consultant to Moderna and Singularity. The remaining authors declare no competing financial or non-financial interests.
