## [Peer Review File · Nature Communications]

Unraveling the molecular architecture of autoimmune thyroid diseases at spatial resolutionEditorial note: Parts of this Peer Review File have been redacted as indicated to remove third-party material where no permission to publish could be obtained.

REVIEWER COMMENTS

Reviewer #1 (expert in spatial transcriptomics):

Martinez-Hernandez applied spatial transcriptomics (10X Visium) on 8 tissue samples spanning normal adjacent thyroid (n = 2), Graves' disease (n = 3), and Hashimoto's thyroiditis (n = 3). They use this data to identify locations of major cell types in each condition and differential expression analyses to identify transcriptional changes unique to each condition. These analyses uncovered a thyroid follicular cell (TFC) subpopulation in both GD and HT (collectively grouped as autoimmune thyroid disease, or AITD) that upregulates CD74/MIF, which potentially are closer to thyroid infiltrating lymphocytes (TILs) in both AITD conditions. The authors further identify the presence of fibroblast and endothelial cell (EC) populations in both conditions. Specifically, they identify two fibroblast subpopulations: myofibroblasts and inflammatory associated fibroblasts (IAFs). They found that GD myofibroblasts also co-expressed ADIRF. Among ECs, they focus on PLVAP+ ECs detected in both GD and HT. They validate these cells with either IHC or IF staining to confirm their locations in the respective AITD tissues. Focusing on the strengths of the paper, the Visium dataset appears high-quality and rich in terms of number genes/spot and other general QC metrics. My main criticism of this manuscript is the lack of spatial insights or use of the spatial data to conduct analyses that are unique to this form of data that otherwise could not be conducted (e.g. with scRNA-seq alone or other methods). Other than determining that CD74/MIF+ TFCs may be closer to TILs, no other spatial relationship is described, which results in a missed opportunity to investigate how some of these transcriptional changes might be occurring at the molecular level (for example, through cell-cell interactions of neighboring cells). Thus, other than a catalogue of some of these cell types with protein validation, it's difficult to connect the findings to advances in the knowledge of the pathogenesis of AITD, and the authors failed to persuade me of the potential value of this dataset. The Discussion focuses on prior studies of these diseases, but in my opinion, draws thin connections to the current study other than highlighting that they found similar cell

types previously discovered. I believe authors could strengthen the manuscript through additional analyses. I detail suggestions and comments below.

Major comments:

1. Figure 1A, need to label which samples are GD, HT, and controls, respectively.
2. While the approach of annotating the samples histologically is reasonable, the data is unbiased in nature and therefore, unbiased data analysis should also be applied. This is important for assessing whether clustering resembles histological annotations. It also helps to assess additional QC concerns such as how much diffusion of transcripts there might be if there is mismatch between histology and clustering. Or it will reveal potential cell types that may be “hidden” to the pathologists’ annotation. The authors do use this approach after the histological annotations (for example, subclustering of TFC spots), but an unbiased initial analysis is important for the above reasons.
3. As mentioned, the authors’ approach relies on histological annotations to guide pseudobulk comparisons across conditions. This approach largely ignores the spatial relationships of different cell types to one another, which is the main advantage of the ST approach. Many studies use this type of data to conduct cell-cell interaction (CCI) inference, which is absent from this study. If such spatial aspects are overlooked, the resulting analysis is similar to laser-capture microdissection with bulk RNA-seq (where prior knowledge of areas is required but not necessarily connected to non-sequenced areas). Why didn't authors attempt these analysis approaches?
4. I commend the authors’ use of a published scRNA-seq of HT (Zhang et al. Nature Comms 2022) to compare with their paper, and they additionally cite a published scRNA-seq of GD (Alvarez-Sierra, Journal of Autoimmunity 2023). Why not use the GD scRNA-seq data to further compare their data?
5. Authors should attempt to use latest deconvolution methods using scRNA-seq data as a reference for Visium data (e.g., cell2location, Nature Biotech 2021). If HT is the only scRNA-seq dataset available, then applying deconvolution to HT samples may help strengthen some conclusions, such as detecting the presence of B/plasma cells in TFC cluster 1 (line 287). However, the GD scRNA-seq is published and could be attempted (as referenced in comment #4). These approaches may better localize cells such as macrophages, T cell subsets, etc. which have been identified in scRNA-seq studies.

6. Figure 7 diagrams some working models of GD and HT pathogenesis which contain several aspects not directly analyzed in the manuscript. For example, macrophages are prominently displayed as a contributors to disease, but macrophages are not analyzed at all within the manuscript. TGF- β is shown as an inducer of myofibroblasts, but no interrogation of TGF- β signaling is shown. Both macrophages and TGF- β signaling could be investigated with this dataset, but authors curiously did not attempt to do either.

7. The Discussion is very long with many references to prior studies and many molecular mechanisms highlighted, such as fibroblast production of IL6, IL11, and IL16 (496), TGF- β induction of myofibroblasts (line 506), chemotaxis through CXCL12 (lines 537). These are all factors that could have been analyzed within the Visium dataset (or in combination with scRNA-seq data), and could have shed additional insight into the pathogenesis, but authors did not discuss how their data potentially support or extend these prior studies. In sum, there seems to be the potential for many different avenues of additional investigation that are largely ignored.

Minor comments:

1. Were control thyroid tissues patient-matched to the GD or HT samples or from independent donors? Please include some donor info either in Fig. 1 or in supplements such as age, gender, treatment status, etc.
2. Some language issues as noted below:
 - a. Line 244: “clusters” implies data analysis; given this refers to histological annotation, I would recommend using the word “regions” instead.
 - b. Line 254, this sentence is not interpretable: “being TILs and GD exclusively in AITD spots and then separated from TFCs and stroma, as expected (Figure 1C, D)”
 - c. Line 300, “being” again is used inappropriately

Reviewer #2 (expert in thyroid autoimmunity):

Nature Communication-Review

The manuscript entitled “Unraveling the molecular architecture of autoimmune thyroid disease at spatial resolution” by Martinez-Hernandez R, de la Blanca NS, Sacristan-Gomez P, et al. described spatial transcriptomics in Graves’ disease (GD) and Hashimoto’s thyroiditis (HT) compared with control thyroid tissues. They have used multiple approaches to analyze tissue transcriptomics for understanding cellular phenotypes including thyroid follicular epithelial cells, connective cells such as fibroblast variants and vascular architecture related cells. Their qualities were also evaluated by multiple approaches such as clustering, differential expression, enrichment, single cell analysis, tissue microarray, IHC and IF. Their findings demonstrated that follicular thyrocytes, fibroblasts, and vascular architectures were different between GD and HT compared to controls. Although pathologies and triggering factors are different in these two diseases, their root causes remain elusive. To identify some essential findings in understanding disease pathogenesis at tissue level, this approach is unique. Some concerns still exist as below:

1. The graphical abstract is too big. It is not necessary to show real images of staining and methodologies, instead they can elaborate by directly writing with a few images as illustrated in the graphical abstract. This will include results as well.
2. Levels of antibody titers are always correlated with the thyroid damage and interestingly these antibodies seem to appear early before symptoms. Did authors identify any

correlations between spatial molecular findings? They need to clarify antibody levels and T3, TSH and T4 levels and their association with transcriptomes.

3. Parameters of lymphocyte phenotypes may give certain association need to be addressed.

Reviewer #3 (expert in thyroid autoimmunity):

In their study, Martínez-Hernández et al. aimed to reveal the molecular and cellular heterogeneity of autoimmune thyroid diseases (AITD). To achieve this, they performed spatial RNAseq (ST) of thyroid sections from six AITD and two control thyroid samples. The results were validated using IHC staining of tissue microarrays consisting of 52 AITD and 23 control thyroid tissues. Compared to previous study on single-cell RNAseq (Zhang et al., 2022) that mainly referred to the immune infiltrating cells, Martínez-Hernández et al, focused mainly on thyroid follicular cells and stromal cells (endothelial cells, fibroblasts). The study is the first ST analysis of AITD and provides several important novel findings, such as identification of diaphragmatic fenestrated vessels that express PLVAP, as well as novel populations of thyroid follicular cells and fibroblasts. Furthermore, these novel data support at the molecular level the possible causative associations between AITD and thyroid cancer that correspond with previous studies on the increased risk of thyroid cancer among AITD patients (PMID: 35903279, PMID: 24084773, PMID: 28443243).

Several issues require comments from authors:

Methodological issues:

1. Unfortunately, patients data are inadequately described. The section "Study subjects" does not provide any information on number of patients and does not refer to their clinical data. The section "Sample preparation and sequencing" mentions "40 fresh-frozen thyroid tissues (21 controls, 11 HT and 8 GD samples)."; however it is not clear where clinical data of these patients are presented. Supplementary file "465982_0_supp_631335_s3p3gq.xls" shows "Visium samples" that include 8 samples subjected to ST, and TMAs table with 23 controls, 28 HTs and 24 GDs that refer to the section "Tissue Microarrays and Immunohistochemistry (IHC) /Immunofluorescence (IF) staining ". It is not clear which (and if) of the "Visium samples" correspond to TMA samples (patients are coded only in Visium samples – lack of patients numbering in TMA samples). In TMA table, patients sex is indicated by 0/1 – please change to F/M. Please clearly indicate which samples were subjected to spatial RNAseq. Please also correct the units provided in the legend of TMA table.

2. Control patients: The information on the diagnoses of control patients is missing (there is only information that the controls consisted of “histologically normal parenchyma adjacent to other thyroid lesions from patients who underwent thyroidectomy” (line 106-17 in the manuscript). Please provide clear information on the clinical diagnoses of control patients. In the supplementary TMA table, several controls lack both T4 and TSH results – how it was assured that they were euthyroid? One control patient lack accurate T4/TSH values (‘normal’ is given). Four control patients have improper hormone levels: one control patient has T4 above normal range (2.76) while another has increased TSH (5.42). Another one has both T4 (1.06) and TSH (0.02) below their normal range.

3. It is not clear how and if and how the patients were treated (e.g. if they received T4 supplementation (and in what form, for how long) or any other medications).

4. The authors state that "Serum free thyroxine 4 (FT4), TSH, and levels of antibodies against thyroglobulin (TG), thyroperoxidase (TPO) and thyrotropin receptor (TSH-R) were measured as previously described 29." However, compared to what is presented in Supplementary Table 1, reference 29 provides different reference ranges for anti-TG and anti-TPO antibodies (in the cited article normal ranges are: anti-Th Ab <40IU/ml, anti-TPO Ab: <25 IU/ml).

5. Section “Sample preparation and sequencing”: please provide detailed procedures (e.g. in Supplement)

6. Supplementary Table 2: please provide catalogue numbers for antibodies

Study findings:

1. The novel data on TFC and stromal cells are definitely of great importance. However, since AITD is mainly the result of immune infiltration, some basic comments on the identified immune cells should be provided (e.g. B lymphocytes, dendritic cells, or Th1, Th2, or Th17 subtypes – the key players in AITD immunology).

2. One HT patient was male: did the authors observe any sex-related differences when compared with female patients?
3. Recent study revealed that CCL21+ myofibroblasts and CCL21+ fibroblasts, contribute to the thyroidal tissue microenvironment in HT (Zhang et al Nat Commun 2022 Feb 9;13(1):775). Did the authors detect these cells in their thyroid sections?
4. The study finds ETC alterations in AITD. How these findings correspond with ETC I deficiency in HT described in 2016 by Zimmermann et al? (paper in Mitochondrion).
5. Regarding increased CD74 in AITD: CD74 is overexpressed and plays oncogenic role in PTC (Cheng et al, Endocrine-Related Cancer, 2015). It also regulates mitochondria dynamics (De et al J Biol Chem. 2018). Could this be another hint for the premalignant state of the analyzed TFCs?
6. Please discuss more the role of the two thyrocyte subpopulations found in HT.
7. Line 371: „We plotted main genes individually to further characterize these subpopulations at protein level”. Not clear; please explain how the protein level was achieved
8. Considering low number of patients analyzed in ST: it needs to be discussed to what extent these data are representative for AITD
9. The authors state that they aimed “to shed light on other therapeutically relevant issues related to heterogeneity such as the interplay between TFCs, connective tissue and vascular cells” (line 461-463). Please discuss more specifically how the presented data can improve current therapeutic options in AITD. How these findings relate to the current therapeutic strategies?
10. The limitations of the study are not properly discussed. Please discuss this issue.

Language/grammar/spelling:

The manuscript is properly written, with minimal spelling errors (e.g. line 258: “;”). Some other issues require corrections:

1. Line 22: affiliation 1 lacks city and country

2. Line 61: “Graves’ 61 ophthalmopathy”: please use term “Thyroid eye disease” that was adopted as more accurate by the leading endocrine societies (Davies, Thyroid. 2022 Dec;32(12):1434-1436; <https://www.thyroid.org/thyroid-eye-disease/>, <https://www.endocrine.org/patient-engagement/endocrine-library/thyroid-eye-disease>)

3. Line 472: "This hypothesis is in line ..." It is not clear what hypothesis the authors have in mind.

Response to Reviewers:

We would sincerely appreciate if the reviewers could reconsider our manuscript after addressing the reviewers' comments. We also thank the reviewers for their helpful comments and the huge amount of work dedicated to review our manuscript and all the data, including the Visium dataset and clinical data of each patient. We have now thoroughly revised spatial and clinical data of all patients included in the cohort and addressed new analyses to strengthen the manuscript.

Reviewer #1 (expert in spatial transcriptomics):

Martinez-Hernandez applied spatial transcriptomics (10X Visium) on 8 tissue samples spanning normal adjacent thyroid (n = 2), Graves' disease (n = 3), and Hashimoto's thyroiditis (n = 3). They use this data to identify locations of major cell types in each condition and differential expression analyses to identify transcriptional changes unique to each condition. These analyses uncovered a thyroid follicular cell (TFC) subpopulation in both GD and HT (collectively grouped as autoimmune thyroid disease, or AITD) that upregulates CD74/MIF, which potentially are closer to thyroid infiltrating lymphocytes (TILs) in both AITD conditions. The authors further identify the presence of fibroblast and endothelial cell (EC) populations in both conditions. Specifically, they identify two fibroblast subpopulations: myofibroblasts and inflammatory associated fibroblasts (IAFs). They found that GD myofibroblasts also co-expressed ADIRF. Among ECs, they focus on PLVAP+ ECs detected in both GD and HT. They validate these cells with either IHC or IF staining to confirm their locations in the respective AITD tissues. Focusing on the strengths of the paper, the Visium dataset appears high-quality and rich in terms of number genes/spot and other general QC metrics. My main criticism of this manuscript is the lack of spatial insights or use of the spatial data to conduct analyses that are unique to this form of data that otherwise could not be conducted (e.g. with scRNA-seq alone or other methods). Other than determining that CD74/MIF+ TFCs may be closer to TILs, no other spatial relationship is described, which results in a missed opportunity to investigate how some of these transcriptional changes might be occurring at the molecular level (for example, through cell-cell interactions of neighboring cells). Thus, other than a catalogue of some of these cell types with protein validation, it's difficult to connect the findings to advances in the knowledge of the pathogenesis of AITD, and the authors failed to persuade me of the potential value of this dataset. The Discussion focuses on prior studies of these diseases, but in my opinion, draws thin connections to the current study other than highlighting that they found similar cell types previously discovered. I believe authors could strengthen the manuscript through additional analyses. I detail suggestions and comments below.

Major comments:

- 1. Figure 1A, need to label which samples are GD, HT, and controls, respectively.**

RESPONSE: *We sincerely appreciate the reviewer for the comments to improve the quality of our work. We have modified Figure 1 and have labeled the corresponding names in each sample.*

2. While the approach of annotating the samples histologically is reasonable, the data is unbiased in nature and therefore, unbiased data analysis should also be applied. This is important for assessing whether clustering resembles histological annotations. It also helps to assess additional QC concerns such as how much diffusion of transcripts there might be if there is mismatch between histology and clustering. Or it will reveal potential cell types that may be “hidden” to the pathologists’ annotation. The authors do use this approach after the histological annotations (for example, subclustering of TFC spots), but an unbiased initial analysis is important for the above reasons.

RESPONSE: *We completely agree with the reviewer's suggestion. In this regard, we first performed an unsupervised analysis and clustering of the spatial data to assess the different patterns within the data and then we compared our results with a histological classification performed by a pathologist. We have included a new Supplementary Figure 4 in the revised version of the manuscript to clarify both analyses performed in this study. As both approaches gave very similar results, we chose the histological annotation instead of the unsupervised analysis to avoid mixed spots and borderline areas.*

3. As mentioned, the authors’ approach relies on histological annotations to guide pseudobulk comparisons across conditions. This approach largely ignores the spatial relationships of different cell types to one another, which is the main advantage of the ST approach. Many studies use this type of data to conduct cell-cell interaction (CCI) inference, which is absent from this study. If such spatial aspects are overlooked, the resulting analysis is similar to laser-capture microdissection with bulk RNA-seq (where prior knowledge of areas is required but not necessarily connected to non-sequenced areas). Why didn't authors attempt these analysis approaches?

RESPONSE: *We thank the reviewer for this appreciation. As previously explained, we decided to follow the histological approach using the spatial information of the data. Using this approach, we could correlate some of our findings such as CD74 positive thyrocytes close to the immune infiltrates (Figure 2) or myofibroblasts surrounding thyroid follicles (Figure 3). In this regard, we also agree that other analyses such as cell-cell interaction could also be performed on spatial data. We have included a new section with cell-cell interaction and deconvolution analyses. Results of each approach have been included in new Figures 2, 3, 5, and 7 and Supplementary figures 6, 7, 9 and 13.*

4. I commend the authors' use of a published scRNA-seq of HT (Zhang et al. Nature Comms 2022) to compare with their paper, and they additionally cite a published scRNA-seq of GD (Alvarez-Sierra, Journal of Autoimmunity 2023). Why not use the GD scRNA-seq data to further compare their data?

RESPONSE: *We thank the reviewer for this suggestion. As mentioned, these data could be a great opportunity to validate our GD data. Unfortunately, raw data of (Álvarez-Sierra D, et al. Single cell transcriptomic analysis of Graves' disease thyroid glands reveals the broad immunoregulatory potential of thyroid follicular and stromal cells and implies a major re-interpretation of the role of aberrant HLA class II expression in autoimmunity. J Autoimmun. 2023 Sep;139:103072. doi: 10.1016/j.jaut.2023.103072) were not publicly available at the time we sent our manuscript to Nature Communications. In this regard, we contacted the corresponding author and, in December 2023, he shared the data in the Sequence Read Archive (SRA) associated with another publication (Mosteiro L, et al. Notch signaling in thyrocytes is essential for adult thyroid function and mammalian homeostasis. Nat Metab. 2023 Dec;5(12):2094-2110. doi: 10.1038/s42255-023-00937-1). We are now analyzing this dataset. Unfortunately, this scRNAseq was performed using cultured thyroid cells derived from tissue surgery samples from GD patients, after the removal of CD45+ immune cells. Thus, this dataset is not suitable to deconvolute our results. Moreover, some of the markers are partially lost in vitro (figure below), thus these results are probably not comparable with our ST data.*

[figure redacted]
[figure redacted]

5. Authors should attempt to use latest deconvolution methods using scRNA-seq data as a reference for Visium data (e.g., cell2location, Nature Biotech 2021). If HT is the only scRNA-seq dataset available, then applying deconvolution to HT samples may help strengthen some conclusions, such as detecting the presence of B/plasma cells in TFC cluster 1 (line 287). However, the GD scRNA-seq is published and could be attempted (as referenced in comment #4). These approaches may better localize cells such as macrophages, T cell subsets, etc. which have been identified in scRNA-seq studies.

RESPONSE: *Following the Referee's suggestions we have used cell2location to deconvolve our spatial transcriptomics using scRNA-seq data from HT (Nature Communications). These data have been included in the revised version of the manuscript. Also, the Results section in the new version of the manuscript has been modified, including a description of the distribution of thyroid infiltrating cells obtained applying deconvolution methods (new Figure 7 and supplementary Figures 6 and 7).*

6. Figure 7 diagrams some working models of GD and HT pathogenesis which contain several aspects not directly analyzed in the manuscript. For example, macrophages are prominently displayed as a contributors to disease, but macrophages are not analyzed at all within the manuscript. TGF-b is shown as an inducer of myofibroblasts, but no interrogation of TGF-b signaling is shown. Both macrophages and TGF-b signaling could be investigated with this dataset, but authors curiously did not attempt to do either.

RESPONSE: We thank the reviewer for this suggestion, and we agree that characterization of macrophages could be important in this study. In this regard, we have also included a new Figure 7 describing the distribution of immune infiltrating cells including myeloid cells (DCs and macrophages could not be distinguished from each other), as described in our analysis of the HT scRNAseq data in Supp Figure 3. Following the reviewer's suggestion, we investigated TGF-beta

in our dataset, and indeed, its expression, as well as its interactions (Figures below), are increased in HT and GD samples. The function of TGF-beta has been widely described in AITD; thus, we have not included these results in the main manuscript.

7. The Discussion is very long with many references to prior studies and many molecular mechanisms highlighted, such as fibroblast production of IL6, IL11, and IL16 (496), TGF-b induction of myofibroblasts (line 506), chemotaxis through CXCL12 (lines 537). These are all factors that could have been analyzed within the Visium dataset (or in combination with scRNA-seq data), and could have shed additional insight into the pathogenesis, but authors did not discuss how their data potentially support or extend these prior studies. In sum, there seems to be the potential for many different avenues of additional investigation that are largely ignored.

RESPONSE: *As stated below, and following the reviewer's suggestion, we have analyzed all these mechanisms in our spatial data. We analyzed the cytokines that have been involved in chemotaxis and in myofibroblast function and found significant results in our data analysis for soluble mediators such as TGF-beta, IL-16, CXCL12 and CCL21. Considering that these mechanisms have been previously validated and due to the huge length of the manuscript, we made the decision not to include them in order to maintain readability and focus. We believe that by doing so, we enhance the clarity and coherence of the paper while still addressing the main points raised by our results.*

Minor comments:

1. Were control thyroid tissues patient-matched to the GD or HT samples or from independent donors? Please include some donor info either in Fig. 1 or in supplements such as age, gender, treatment status, etc.

RESPONSE: *Control thyroid samples included in ST data were obtained from healthy thyroid tissue from independent donors with non-autoimmune thyroid tissue. Clinical information has been completed in Supplementary Table 1.*

2. Some language issues as noted below:

a. Line 244: "clusters" implies data analysis; given this refers to histological annotation, I would recommend using the word "regions" instead.

RESPONSE: *We have changed "clusters" to "regions" when referring to the histological annotation.*

b. Line 254, this sentence is not interpretable: "being TILs and GD exclusively in AITD spots and then separated from TFCs and stroma, as expected (Figure 1C, D)"

RESPONSE: *We apologize to the reviewer for the inconveniences and thank you for the appreciation. There was a typo in that phrase: We referred to TILs and GC (germinal center). We have modified the sentence and corrected the typo.*

c. Line 300, "being" again is used inappropriately

RESPONSE: *Following reviewer' suggestions we have corrected the corresponding language editing issues in the new version of the manuscript.*

Reviewer #2 (expert in thyroid autoimmunity):

The manuscript entitled "Unraveling the molecular architecture of autoimmune thyroid disease at spatial resolution" by Martinez-Hernandez R, de la Blanca NS, Sacristan-Gomez P, et al. described spa.al transcriptomics in Graves' disease (GD) and Hashimoto's thyroiditis (HT) compared with control thyroid tissues. They have used multiple approaches to analyze tissue transcriptomics for understanding cellular phenotypes including thyroid

follicular epithelial cells, connective cells such as fibroblast variants and vascular architecture related cells. Their qualities were also evaluated by multiple approaches such as clustering, differential expression, enrichment, single cell analysis, tissue microarray, IHC and IF. Their findings demonstrated that follicular thyrocytes, fibroblasts, and vascular architectures were different between GD and HT compared to controls. Although pathologies and triggering factors are different in these two diseases, their root causes remain elusive. To identify some essential findings in understanding disease pathogenesis at tissue level, this approach is unique. Some concerns still exist as below:

1. The graphical abstract is too big. It is not necessary to show real images of staining and methodologies, instead they can elaborate by directly writing with a few images as illustrated in the graphical abstract. This will include results as well.

RESPONSE: *We thank the reviewer for the suggestion. We have modified the graphical abstract in order to give a clearer and more precise message of our study. We have included the graphical abstract in Figure 1A.*

2. Levels of antibody titers are always correlated with the thyroid damage and interestingly these antibodies seem to appear early before symptoms. Did authors identify any correlations between spatial molecular findings? They need to clarify antibody levels and T3, TSH and T4 levels and their association with transcriptomes.

RESPONSE: *We thank the reviewer for this suggestion. We have included a new Figure 4 (correlation plots) depicting the correlations of those markers quantified by IHC with hormone levels and antibody titers of the patients. Additionally, we also provide to the reviewer a correlation map of clinical data (hormones and antibody titers) with our molecular findings (see below). However, due to the reduced number of samples and hence the lack of statistical power, we have decided not to include these last plots in the new version of the manuscript.*

3. Parameters of lymphocyte phenotypes may give certain association need to be addressed.

RESPONSE: We thank the reviewer for this suggestion. We have performed an analysis of the spatial distribution of thyroid infiltrating immune cells using spot-deconvolution tools. The results of the different subpopulations of immune cells are included in a new Figure 7. The Results section has been modified in the new manuscript to describe these results.

Reviewer #3 (expert in thyroid autoimmunity):

In their study, Martínez-Hernández et al. aimed to reveal the molecular and cellular heterogeneity of autoimmune thyroid diseases (AITD). To achieve this, they performed spatial RNAseq (ST) of thyroid sections from six AITD and two control thyroid samples. The results were validated using IHC staining of tissue microarrays consisting of 52 AITD and 23 control thyroid tissues. Compared to previous study on single-cell RNAseq (Zhang et al., 2022) that mainly referred to the immune infiltrating cells, Martínez-Hernández et al, focused mainly on thyroid follicular cells and stromal cells (endothelial cells, fibroblasts). The study is the first ST analysis of AITD and provides several important novel findings, such as identification of diaphragmatic fenestrated vessels that express PLVAP, as well as novel populations of thyroid follicular cells and fibroblasts. Furthermore, these novel data support at the molecular level the possible causative associations between AITD and

thyroid cancer that correspond with previous studies on the increased risk of thyroid cancer among AITD patients (PMID: 35903279, PMID: 24084773, PMID: 28443243).

Several issues require comments from authors:

Methodological issues:

1. Unfortunately, patients data are inadequately described. The section "Study subjects" does not provide any information on number of patients and does not refer to their clinical data. The section "Sample preparation and sequencing" mentions "40 fresh-frozen thyroid tissues (21 controls, 11 HT and 8 GD samples)."; however it is not clear where clinical data of these patients are presented. Supplementary file "465982_0_supp_631335_s3p3gq.xls" shows "Visium samples" that include 8 samples subjected to ST, and TMAs table with 23 controls, 28 HTs and 24 GDs that refer to the section "Tissue Microarrays and Immunohistochemistry (IHC) /Immunofluorescence (IF) staining ". It is not clear which (and if) of the "Visium samples" correspond to TMA samples (patients are coded only in Visium samples – lack of patients numbering in TMA samples). In TMA table, patients sex is indicated by 0/1 – please change to F/M. Please clearly indicate which samples were subjected to spatial RNAseq. Please also correct the units provided in the legend of TMA table.

RESPONSE: *We apologize for the lack of some information regarding the description of the patient data. We have revised Supplementary Table 1 and included those patients corresponding to the samples used in the transcriptome analysis. Additionally, we have incorporated clinical data of patients undergoing treatment and pathological data of control samples.*

The histology and RIN quality of 40 fresh-frozen tissues were assessed, and eight samples, representative of each condition (control, HT or GD) with RIN>7 were selected for ST analysis. To prevent misinterpretation of these samples, we have removed this sentence and now only mention the eight samples used in the ST analysis.

2. **Control patients:** The information on the diagnoses of control patients is missing (there is only information that the controls consisted of "histologically normal parenchyma adjacent to other thyroid lesions from patients who underwent thyroidectomy" (line 106-17 in the manuscript). Please provide clear information on the clinical diagnoses of control patients. In the supplementary TMA table, several controls lack both T4 and TSH results – how it was assured that they were euthyroid? One control patient lack accurate T4/TSH values ('normal' is given). Four control patients have improper hormone levels: one control patient has T4 above normal range (2.76) while another has increased TSH (5.42). Another one has both T4 (1.06) and TSH (0.02) below their normal range.

RESPONSE: *We appreciate the reviewer's insightful observations. Control samples were meticulously chosen by an expert in thyroid pathology, specifically from patients who underwent thyroidectomy with diagnoses of microcarcinomas, follicular/Hürthle adenoma, or localized*

multinodular goiter (all samples exhibited histologically normal parenchyma). Supplementary Table 1 now includes additional information about the clinical diagnoses of these control patients. None of these patients exhibited positive antibody titers, ruling out thyroid autoimmunity. Hormone values were within normal levels for the majority of patients. We conducted a thorough review of the analytics and identified and corrected several mistakes in the revised version.

3. It is not clear how and if and how the patients were treated (e.g. if they received T4 supplementation (and in what form, for how long) or any other medications).

RESPONSE: *We thank the reviewer for these observations. We have included the treatment of all patients before surgery in Supplementary Table 1.*

4. The authors state that "Serum free thyroxine 4 (FT4), TSH, and levels of antibodies against thyroglobulin (TG), thyroperoxidase (TPO) and thyrotropin receptor (TSH-R) were measured as previously described 29." However, compared to what is presented in Supplementary Table 1, reference 29 provides different reference ranges for anti-TG and anti-TPO antibodies (in the cited article normal ranges are: anti-Th Ab <40IU/ml, anti-TPO Ab: <25 IU/ml).

RESPONSE: *We apologize for this misquotation. In the revised version of the manuscript, we have described the detection methods used for hormone and antibody titers in the section "Material and Methods: Study subjects".*

5. Section "Sample preparation and sequencing": please provide detailed procedures (e.g. in Supplement)

RESPONSE: *We thank the reviewer for this suggestion. We have included a detailed procedure of the Visium Spatial gene expression protocol in the Section "Material and Methods: Sample preparation and sequencing"*

6. Supplementary Table 2: please provide catalogue numbers for antibodies

RESPONSE: *We thank the reviewer for the comment. We have modified Supplementary Table 2 indicating that catalogue number is equivalent to the Reference of the antibody.*

Study findings:

1. The novel data on TFC and stromal cells are definitely of great importance. However, since AITD is mainly the result of immune infiltration, some basic comments on

the identified immune cells should be provided (e.g. B lymphocytes, dendritic cells, or Th1, Th2, or Th17 subtypes – the key players in AITD immunology).

RESPONSE: *We thank the reviewer for their comments. We have included in the new Results section the results of the transcriptomic analysis of thyroid-infiltrating immune cells (CD4/CD8 T cells, Natural killer-NK, dendritic cells-DC, naive and germinal center associated-B cells-NaB and GCB, macrophages-Mac and plasmablasts-PB), confirming their presence in our samples. Unfortunately, we were not able to generate an informative plot specifically for Th17 cells, as key genes related to Th17 cells, such as ROR-gamma, are not highly detected transcriptomically. Additionally, we have included deconvolution results on these data to confirm our findings.*

2. One HT patient was male: did the authors observe any sex-related differences when compared with female patients?

RESPONSE: *We thank the reviewer for the comment. We believe that there is no batch effect due to the gender, since the integration of the transcriptomic data of Visium samples was homogeneous among HT patients. Indeed, immunohistochemical/immunofluorescence analysis included samples from both genders, and the different stainings analyzed showed similarity between males and females. Overall, the results obtained could be attributed to the disease condition rather than gender.*

3. Recent study revealed that CCL21+ myofibroblasts and CCL21+ fibroblasts, contribute to the thyroidal tissue microenvironment in HT (Zhang et al Nat Commun 2022 Feb 9;13(1):775). Did the authors detect these cells in their thyroid sections?

RESPONSE: *Thanks to the reviewer for the comment. We have detected CCL21 expression in immune infiltrating regions of AITD samples, specifically in those areas surrounding germinal centers (new Figure 7).*

We plotted CCL21 and other cytokines associated with chemotaxis, confirming the presence of CCL21+ cells in our thyroid sections (see figure below). We did not perform immunostaining experiments, as Zhang, et al. already demonstrated the presence of this myofibroblasts/fibroblasts in HT tissue.

4. The study finds ETC alterations in AITD. How these findings correspond with ETC I deficiency in HT described in 2016 by Zimmermann et al? (paper in Mitochondrion)

RESPONSE: *Thanks to the reviewer for the query. Zimmermann, et al. studied the loss of respiratory chain complex I in HT samples with multiple oncocytic lesions. In our samples, we did not observe any sample characterized by oncocytic features. Additionally, Zimmermann et al observed an alteration in ETC complex I attributed to one unique gene (NDUFS4) among the several genes related to this complex (38 subunits encoded in nuclear DNA and seven subunits encoded in mitochondrial DNA). In our ST data, transcriptomic levels of NDUFS4 are not significantly different between TFCs conditions (**Supplementary Data 3**). They also reported an increased mitochondrial mass, which is in concordance with the increase of different genes related to ETC in our analysis. In the revised version of the manuscript, we have mentioned these results and included the reference of the study by Zimmermann et al in the Discussion section.*

5. Regarding increased CD74 in AITD: CD74 is overexpressed and plays oncogenic role in PTC (Cheng at al, Endocrine-Related Cancer, 2015). It also regulates mitochondria dynamics (De at al J Biol Chem. 2018). Could this be another hint for the premalignant state of the analyzed TFCs?

RESPONSE: *We thank the reviewer for this comment. As reported by Cheng, et al., CD74 is associated with tumor progression and malignancy, likely due to its tight relationship with proliferation and survival signaling pathways. Moreover, Anti-CD74 therapies have been tested in vitro in papillary thyroid cancer models, resulting in a global reduction in migration, invasion or proliferation. Moreover, De, et al. demonstrated that the signaling pathway involving MIF-CD74-NFKB promotes mitochondria stabilization and tumor cell expansion. Indeed, silencing of MIF or CD74, or blocking NF-KB signaling, led to alterations in mitochondria and cell death. Considering these results, the expression of CD74 in TFCs from AITD patients might indicate a premalignant state, but further insights are needed to confirm this hypothesis, as CD74 has other functions. CD74 is part of MHC-II complex. This complex is involved in antigen presentation, a key process in the activation of immune cells contributing to the pathogenesis of AITD. Moreover, CD74-MIF play important roles in injury responses. Thus, in the context of AITD with autoantibodies against thyroid antigens and proinflammatory cytokines, TFCs may undergo injury repair via CD74-MIF signaling.*

In conclusion, CD74 and MIF signaling could be related to a premalignant state in TFCs, representing an intriguing area of research, but further evidence is needed to fully elucidate the association of CD74-MIF with tumorigenesis.

6. Please discuss more the role of the two thyrocyte subpopulations found in HT.

RESPONSE: *We appreciate the reviewer's suggestion. In response, we have expanded the discussion elaborating on the potential role of damaged TFCs in tumorigenesis, also including additional references.*

7. Line 371: „We plotted main genes individually to further characterize these subpopulations at protein level”. Not clear; please explain how the protein level was achieved

RESPONSE: *We have rewritten this sentence in the revised version of the manuscript.*

8. Considering low number of patients analyzed in ST: it needs to be discussed to what extent these data are representative for AITD

RESPONSE: *Thanks to the reviewer for the comment. We acknowledge that the number of patients included in Spatial transcriptomics is limited, which is one of the article’s limitations. Only four samples are included per each ST experiment, so we decided to conduct two experiments, reaching a total of eight samples. We included a larger cohort of patients in the ST data validation by IHC/IF to confirm the results obtained by transcriptomics. Additionally, we have added a new section discussing the limitations of the study to clarify the need for further experimental studies to elucidate the potential functional roles of the new data presented in the study.*

9. The authors state that they aimed “to shed light on other therapeutically relevant issues related to heterogeneity such as the interplay between TFCs, connective tissue and vascular cells” (line 461-463). Please discuss more specifically how the presented data can improve current therapeutic options in AITD. How these findings relate to the current therapeutic strategies?

RESPONSE: *AITD is a multifactorial disease that has posed different challenges to the search for an effective treatment, as no current therapy has achieved complete remission. The identification of abnormalities in gene expression or in cellular interactions in the thyroid tissue in these patients may not always be reflected in changes in systemic serum markers such as thyroid hormone levels or the presence of thyroid auto-antibodies. These tissue-specific abnormalities, may for example help clinicians to better understand why some patients continue to experience symptoms despite seemingly adequate treatment. Also, the contribution of specific cell types and/or molecular pathways dysregulated in AITD could identify drivers of the autoimmune response. This knowledge can lead to the development of targeted therapies aimed at modulating these specific pathways. The identification of novel targets may represent opportunities for the development of innovative therapies, including gene therapies, small molecule inhibitors, or biologics such as antibodies against CD74 [Frölich D, et al. The anti-CD74 humanized monoclonal antibody, milatuzumab, which targets the invariant chain of MHC II complexes, alters B-cell proliferation, migration, and adhesion molecule expression. Arthritis Res Ther. 2012 Mar 9;14(2):R54. doi: 10.1186/ar3767; Wallace DJ, et al. Experience with milatuzumab, an anti-CD74 antibody against immunomodulatory macrophage migration inhibitory factor (MIF) receptor, for systemic lupus erythematosus (SLE). Annals of the Rheumatic Diseases 2021;80:954-955] or targeting fibroblasts as a novel therapeutic approach [Nygaard G, Firestein GS. Restoring synovial homeostasis in rheumatoid arthritis by targeting fibroblast-like synoviocytes. Nat Rev*

Rheumatol. 2020 Jun;16(6):316-333. doi: 10.1038/s41584-020-0413-5]. By expanding the repertoire of therapeutic targets, spatial transcriptomics can potentially offer more tailored and effective treatment options for AITDs. The identity of molecular subtypes of AITDs may also provide personalized medicine approaches by predicting individual patient responses to specific treatments. These techniques could be used in the future to analyze gene expression changes in response to treatment, validate the efficacy of existing therapies, and identify potential mechanisms of treatment resistance. This information can guide the optimization of current treatment protocols or the development of combination therapies to enhance clinical outcomes. We have included in the discussion possible therapy applications related to the results of our study.

10. The limitations of the study are not properly discussed. Please discuss this issue.

RESPONSE: *We have included a new section discussing the limitations of our study, focusing on the low number of patients included in Spatial transcriptomics and the need to develop further experimental evidence to clarify the role of the different cell subpopulations found in AITD.*

Language/grammar/spelling:

The manuscript is properly written, with minimal spelling errors (e.g. line 258: “;,”). Some other issues require corrections:

- 1. Line 22: affiliation 1 lacks city and country**
- 2. Line 61: “Graves’ 61 ophthalmopathy”:** please use term “Thyroid eye disease” that was adopted as more accurate by the leading endocrine societies (Davies, *Thyroid. 2022 Dec;32(12):1434-1436;* <https://www.thyroid.org/thyroid-eye-disease/>, <https://www.endocrine.org/patient-engagement/endocrine-library/thyroid-eye-disease>)
- 3. Line 472: "This hypothesis is in line ..."** It is not clear what hypothesis the authors have in mind.

RESPONSE: *Thanks to the reviewer for the kind suggestions. We have corrected the indicated issues in the revised version of the manuscript.*

REVIEWERS' COMMENTS

Reviewer #1 (Remarks to the Author):

I commend authors on this thoroughly revised manuscript that has addressed my concerns. I believe it is suitable for publication.

Reviewer #1 (Remarks on code availability):

I reviewed the scripts for organization, accessibility, and general readability, which are all satisfactory.

Reviewer #2 (Remarks to the Author):

This reviewer was no longer available for review. The responses to their comments were assessed by the editorial office.

Reviewer #3 (Remarks to the Author):

The Authors addressed all issues raised in my review. Congratulations on this very interesting and important study.